# Regulation of localization and function of the transcriptional co-activator YAP by angiomotin

Susana Moleirinho[1†], Sany Hoxha[1†], Vinay Mandati[1], Graziella Curtale[1], Scott Troutman[1], Ursula Ehmer[2], Joseph L Kissil[1*]

[1]Department of Molecular Medicine, The Scripps Research Institute, Jupiter, United States; [2]Department of Medicine II, Klinikum rechts der Isar, Technische Universität München, Munich, Germany

**Abstract** The Hippo-YAP pathway is a central regulator of cell contact inhibition, proliferation and death. There are conflicting reports regarding the role of Angiomotin (Amot) in regulating this pathway. While some studies suggest a YAP-inhibitory function other studies indicate Amot is required for YAP activity. Here, we describe an Amot-dependent complex comprised of Amot, YAP and Merlin. The phosphorylation of Amot at Serine 176 shifts localization of this complex to the plasma membrane, where it associates with the tight-junction proteins Pals1/PATJ and E-cadherin. Conversely, hypophosphorylated Amot shifts localization of the complex to the nucleus, where it facilitates the association of YAP and TEAD, induces transcriptional activation of YAP target genes and promotes YAP-dependent cell proliferation. We propose that phosphorylation of Amot[S176] is a critical post-translational modification that suppresses YAP's ability to promote cell proliferation and tumorigenesis by altering the subcellular localization of an essential YAP co-factor.

*For correspondence: jkissil@ scripps.edu

[†]These authors contributed equally to this work

Competing interests: The authors declare that no competing interests exist.

## Introduction

Angiomotin (Amot) was originally identified as an angiostatin-binding protein involved in the regulation of endothelial cell polarization, migration, proliferation, and angiogenesis (*Kikuno et al., 1999*; *Levchenko et al., 2003*; *Troyanovsky et al., 2001*). Amot is expressed as two isoforms generated by alternative splicing, full-length Amot-p130 (referred to as Amot in this manuscript) and truncated Amot-p80, which lacks the 409-amino acid N-terminus (*Ernkvist et al., 2006*). Amot, Angiomotin-like 1 (AmotL1) and Angiomotin-like 2 (AmotL2) compose the Motin family of proteins, which share an N-terminus with conserved glutamine-rich domains and PPxY motifs, and a C-terminus containing conserved coiled-coil (CC) and PDZ-binding domains (*Bratt et al., 2002*; *Nishimura et al., 2002*; *Zheng et al., 2009*). Although structurally similar, the functions of Motin family members appear to be distinctive, as divergent spatiotemporal and expression levels have been described across human and mouse tissues and cell lines for each of the family members (*Moleirinho et al., 2014*).

Functionally, Amot is required for endothelial cell migration, by binding to the Syx:Patj/Mupp1 polarity complex to localize RhoA activity to the leading edge of migratory cells (*Ernkvist et al., 2009*). It has also been implicated in epithelial cell polarity, as an inhibitor of the GTPase-Activating Protein (GAP) Rich1 and shown to compromise the integrity of tight junctions by promoting Rich1-mediated hydrolysis of Rho GTPases Rac1 and Cdc42 (*Wells et al., 2006*; *Yi et al., 2011*). Amot regulates collective migration of epithelial cells as part of a signaling axis composed of Merlin-Amot-Rich1 and the Rac1 small GTPase (*Das et al., 2015*). At tight junctions (TJ), TJ-associated Merlin inhibits Rac1 activation through the Amot-Rich1 axis, relocalizing Merlin to the cytoplasm during cell migration and releasing Rac1 from a suppressed state (*Das et al., 2015*; *Yi et al., 2011*). Amot also

**eLife digest** Cells in animals and other multi-cellular organisms need to know when and where they should grow and divide. Individual cells communicate with their surrounding environment and each other via signaling pathways such as the Hippo-YAP pathway, which stimulates cells to grow and therefore influences the size of organs. When the Hippo part of the pathway is active it causes a protein known as YAP to move out of a compartment in the cell called the nucleus. Inside the nucleus, YAP helps to activate genes that promote cell growth. If the Hippo pathway can no longer respond to cues from the environment, YAP becomes over-active and can contribute to the development of various cancers. Therefore researchers are trying to better understand how it is regulated.

Many signals both from inside and outside the cell influence YAP activity. For example, some signals block YAP from entering the nucleus, whereas others cause YAP to be broken down entirely. Several studies have recently identified a signal protein called angiomotin as a regulator of YAP. However, the studies provide conflicting reports as to whether angiomotin promotes or inhibits cell growth.

Like many other proteins, angiomotin can be tagged with a small molecule called a phosphate group that can alter its activity. Moleirinho, Hoxha et al. studied human cells containing versions of angiomotin that mimic different forms of the protein with or without the phosphate. The experiments indicate that when a phosphate is attached at a particular position (known as serine 176), angiomotin predominantly interacts with YAP and another protein called Merlin at the cell surface. On the other hand, when angiomotin does not have a phosphate attached to it, all three proteins can move into the nucleus, where YAP is able to activate genes and promote cell growth.

Overall, these findings indicate that adding a phosphate group to angiomotin can act as a switch to regulate where in the cell it and YAP are found and thus, whether YAP is active. Future experiments will investigate which enzymes add the phosphate group to serine 176, and when they are able to do so.

directly binds to YAP, a central effector of the Hippo signaling pathway, via $^{106}$LPTY$^{109}$/$^{239}$PPxY$^{242}$ motifs present in the N-terminal domain of Amot-p130 and WW domain present in YAP (*Yi et al., 2013*, *2011*; *Zhao et al., 2011*).

The Hippo-YAP signaling pathway, originally characterized in *Drosophila*, is highly conserved in mammals and regulates organ size, cell contact inhibition, proliferation, apoptosis and polarity (*Ramos and Camargo, 2012*; *Yi and Kissil, 2010*; *Yu et al., 2015*). At the core of the mammalian pathway, MST1/2 promote phosphorylation of WW45 (also known as Sav1), Lats1/2, and MOB1 (MOBKL1A/B). Once activated, Lats1/2 phosphorylates the primary downstream effector YAP, promoting ubiquitination and proteosomal degradation by a SCF$^{beta-TRCP}$ E3 ligase and sequestering YAP from the nucleus, where it functions as a transcriptional co-activator (*Hao et al., 2008*; *Zhao et al., 2010b*, *2007*). Among the different transcription factors activated by YAP, the DNA-binding TEA domain (TEAD) transcription factors are thought to mediate a YAP-driven pro-proliferative gene expression program (*Galli et al., 2015*; *Zhao et al., 2010b*, *2007*).

YAP's involvement in cancer has been demonstrated in several tissues, including liver, intestine, heart, pancreas, and brain (*Yu et al., 2015*; *Guerrant et al., 2016*). Importantly, recent studies revealed YAP plays a key role in developing resistance to RAF- and MEK-targeted therapies in lung and colon cancer cells (*Lin et al., 2015*) and cancer relapse in *KRAS*-driven colon and pancreatic cancers (*Kapoor et al., 2014*; *Shao et al., 2014*). Upstream of the core kinase cassette is the tumor suppressor Merlin (moesin-ezrin-radixin-like protein), which is inactivated in Neurofibromatosis type 2 (NF2) (*McClatchey et al., 1998*). Functions for Merlin have been described in the nucleus (*Li et al., 2014*, *2010*) as well as at the plasma membrane (*Yin et al., 2013*; *Zhang et al., 2010*).

In the mouse-liver, homozygous deletion of *Nf2* results in tumor formation. However, heterozygous deletion of *Yap* significantly suppresses the loss-of-*Nf2* phenotype, thus implicating YAP as a major downstream effector of NF2 (*Zhang et al., 2010*). Analysis of liver-specific *Nf2* knockout mice and *Nf2:Amot* double knockout (DKO) mice showed Amot is required for hepatic ductal cell

proliferation and tumor formation in the context of either *Nf2* loss or DDC (3,5-diethoxycarbonyl-1,4-dihydrocollidine)-induced injury. Additionally, substantially increased expression of Amot was observed in *NF2*-null human schwannomas samples, which primarily displayed localization of Amot in the nucleus (*Yi et al., 2013*). Further evidence demonstrated that in renal cell carcinoma (RCC), Amot promotes the proliferation of renal epithelial and RCC cells, and is crucial for YAP transcriptional activity by promoting its nuclear localization (*Lv et al., 2016*). However, it has also been reported that Amot functions as a negative regulator of YAP activity, as direct association of Amot and YAP results in translocation to cytoplasm/cell junctions (*Leung and Zernicka-Goetz, 2013*; *Zhao et al., 2011*). Moreover, a number of studies suggest that Amot acts as a scaffold protein to promote localization of YAP, as well as other Hippo/YAP core kinases, to the cytoplasm/cell junctions/actin cytoskeleton and promote Lats1/2-mediated phosphorylation of YAP (*Chan et al., 2011*; *Dai et al., 2013*; *Paramasivam et al., 2011*; *Wang et al., 2011*). One possible explanation for these opposing regulatory functions of Amot could be attributed to post-translational modifications. Phosphorylation of Ser175 on human Amot (corresponding to Ser176 in mouse) at the conserved HVRSLS motif by Lats1/2 has been reported as a key post-translational modification controlling Amot function and association with YAP (*Hirate et al., 2013*; *Moleirinho et al., 2014*). Here we show that Amot-p130 functions as a scaffold protein mediating YAP and Merlin association and that phosphorylation of Amot at Ser176 induces translocation of the complex from the cytoplasm and nucleus to the plasma membrane. This relocalization of the Amot/YAP/Merlin complex significantly impacts the activity of YAP and regulates YAP's ability to promote cell proliferation and tumorigenesis. These results uncover an unrecognized layer of regulation in the Hippo-YAP pathway and resolve questions regarding the seemingly opposing functions of Amot.

## Results

### Amot, YAP and Merlin form a complex that localizes to the nucleus and cytoplasm

Previous work by multiple groups identified the p130 splice isoform of Angiomotin (Amot) as a binding partner of YAP and Merlin (*Chan et al., 2011*; *Yi et al., 2013*, *2011*; *Zhao et al., 2011*). We first thought to assess whether Amot-p130, YAP and Merlin form a complex and subsequently whether this is dependent on Angiomotin phosphorylation. We used HEK293, as they express relatively high levels of endogenous Angiomotin. First, to determine the interaction between Merlin and YAP, HEK293 cells were transfected with expression vectors for HA-tagged Merlin and V5-tagged YAP. Reciprocal co-immunoprecipitation (co-IP) analyses using anti-HA or anti-V5 antibodies revealed association between YAP and Merlin (*Figure 1A*). The Merlin-YAP interaction was also confirmed in another independent cell type – the hSC2λ immortalized human Schwann cells (*Figure 1B*). Importantly, further validation of the association between YAP and Merlin was confirmed at the endogenous protein levels, using antibodies directed against Merlin or YAP (*Figure 1C*).

   As previous studies reported multiple cellular localizations for Merlin and YAP, we sought to determine the cellular compartments where YAP and Merlin co-localize. Flag-tagged Merlin and V5-tagged YAP were co-transfected in both HEK293 and hSC2λ cells and the localization and association of these proteins were examined by subcellular fractionation coupled to IP with an anti-Flag antibody. These subcellular fractionation studies showed an association between Merlin and YAP primarily in the nucleus and cytoplasm, and to a much lesser extent at the plasma membrane (*Figure 1D–E*). Further confirmation of Merlin and YAP co-localization was provided by immunofluorescence (IF) staining, in HEK293 and hSC2λ cells, where we observed co-localization of IF signal predominantly in the cytoplasm and nucleus (*Figure 1F*). Thus, we conclude that YAP associates with Merlin, co-localizing mostly in the cytoplasm and nucleus.

### YAP-Merlin complex formation is dependent on Amot-p130

Since Amot directly binds YAP and Merlin through two distinct domains (*Yi et al., 2013*, *2011*), we hypothesized that Amot could act as a scaffold and recruit YAP and Merlin into the same complex. To confirm this hypothesis we generated HEK293 cells stably expressing a lentiviral short hairpin RNA specific to Amot (HEK293-shAmot cells). Co-IP experiments with HA-tagged Merlin and V5-tagged YAP confirmed that in the absence of Amot, the complex is not formed as YAP and Merlin

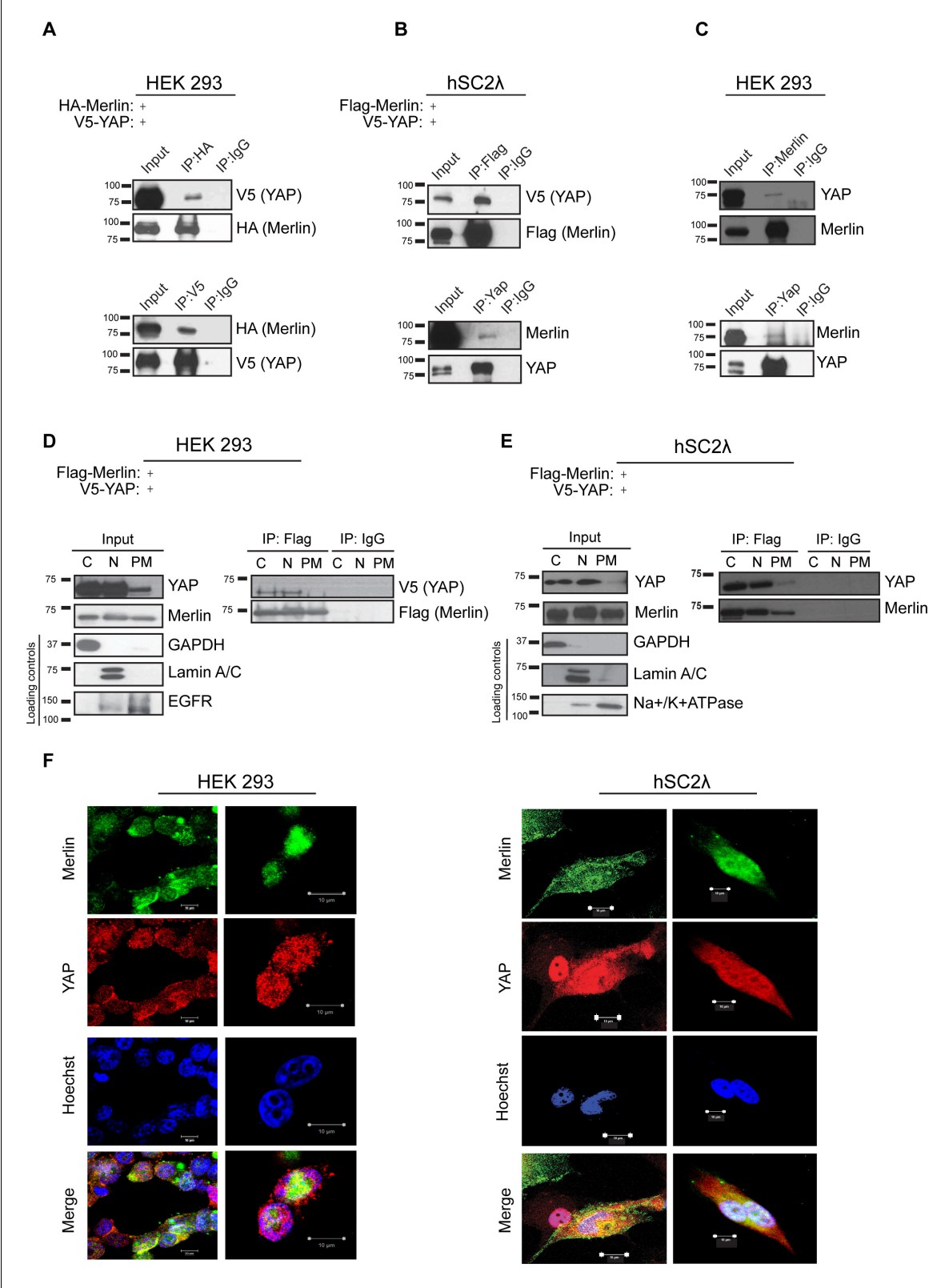

**Figure 1.** YAP associates with Merlin in the nucleus and cytoplasm. Co-immunoprecipitation of YAP and Merlin. (**A**) HEK293 or (**B**) hSC2λ cells were co-transfected with expression plasmids for HA-Merlin or Flag-Merlin and V5-YAP. Total cell lysates (Input) and HA, Flag, YAP or V5 immunoprecipitates (IP) were subjected to immunoblotting analysis with anti-V5, anti-HA, anti-Flag, Merlin or YAP antibodies as indicated. (**C**) Association of endogenous YAP and Merlin. Total lysates from HEK293 cells (input) or IPs with anti-Merlin or anti-YAP antibodies were subjected to immunoblotting analysis with

*Figure 1 continued on next page*

*Figure 1 continued*

indicated antibodies. (D–F) YAP associates with Merlin in the nucleus and in the cytoplasm. (D) HEK293 or (E) hSC2λ cells expressing Flag-Merlin and V5-YAP were fractionated into cytoplasmic, nuclear, and plasma membrane fractions. Cell lysates (input) and V5-IP or Flag-IP of each subcellular fraction were subjected to immunoblot analysis with indicated antibodies. GAPDH, Lamin A/C, and EGFR/ Na$^+$/K$^+$ATPase were used as fractionation controls for the cytoplasmic, nuclear, and plasma membrane fractions, respectively. IgG was used as a non-specific antibody control for IPs throughout. The blots shown are representative of three independent biological replicates (n = 3). (F) HEK293 (left) or hSC2λ (right) cells were co-transfected with YAP and Merlin expression plasmids and subjected to immunofluorescence staining with anti-YAP and anti-Merlin antibodies. Hoechst was used for nuclei fluorescence staining. Pictures show fields at 63x magnification and representative of three independent biological replicates, in each of which 20 independent fields were examined. Scale bar = 10 μm.

could no longer be co-immunoprecipitated (*Figure 2A*). Similarly, knockdown of Amot using an siRNA smartpool impaired the association of Merlin and YAP (*Figure 2—figure supplement 1*).

To further validate the scaffolding role of Amot we evaluated whether Amot-80 can support the interaction between Merlin and YAP. Amot-p80 lacks the 409 amino acid N-terminus present in Amot, which harbors the PPxY motifs ($^{239}$PPEY$^{242}$ and $^{284}$PPEY$^{287}$), and an unconventional LPTY motif ($^{106}$LPTY$^{109}$) that mediate the interaction with YAP (*Ernkvist et al., 2006*; *Wang et al., 2012*; *Yi et al., 2013*) (*Figure 2B*). As expected, attempting to IP HA-Merlin or V5-YAP demonstrated that

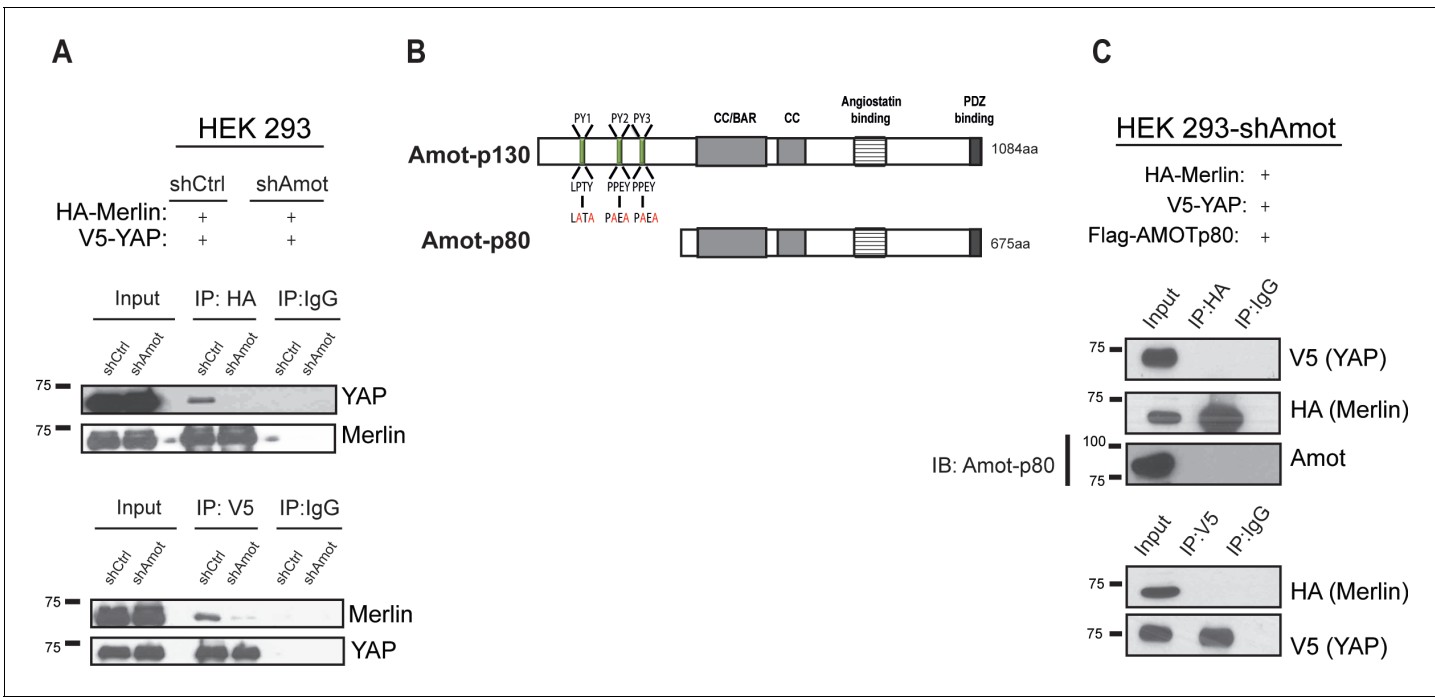

**Figure 2.** YAP and Merlin association is dependent on Amot-p130. (**A**) HEK293 cells stably infected with lentiviral vectors encoding a control shRNA (shCtr) or shRNA targeting Amot (shAmot) were co-transfected with expression plasmids for HA-Merlin and V5-YAP. Total lysates (input) and HA or V5 IPs were subjected to immunoblot analysis with anti-YAP and anti-Merlin antibodies as indicated. (**B**) Graphical representation of Amot p130 and p80 isoforms. Amot-p130 N-terminus PPxY and LPxY motifs are shown, and the generated PY motif mutants are highlighted in red. See *Figure 2—figure supplement 1*. CC/BAR – Coiled-Coil/(Bin/Amphiphysin/Rvs) domain. PDZ – Post synaptic density protein (PSD95, Drosophila disc large (Dlg1) and Zonula occludens-1 (ZO-1) domain. (**C**) HEK293-shAmot cells were co-transfected with HA-Merlin, V5-YAP, and Flag-Amot-p80. Total lysates (Input) and HA or V5 IPs were subjected to immunoblot analysis with anti-V5 and anti-HA antibodies as indicated. The blots shown are representative of three independent biological replicates (n = 3).

The following figure supplements are available for figure 2:

**Figure supplement 1.** YAP and Merlin association is dependent on Amot-p130.

**Figure supplement 2.** YAP/Merlin complex require Amot-p130 PPxY and LPxY motifs.

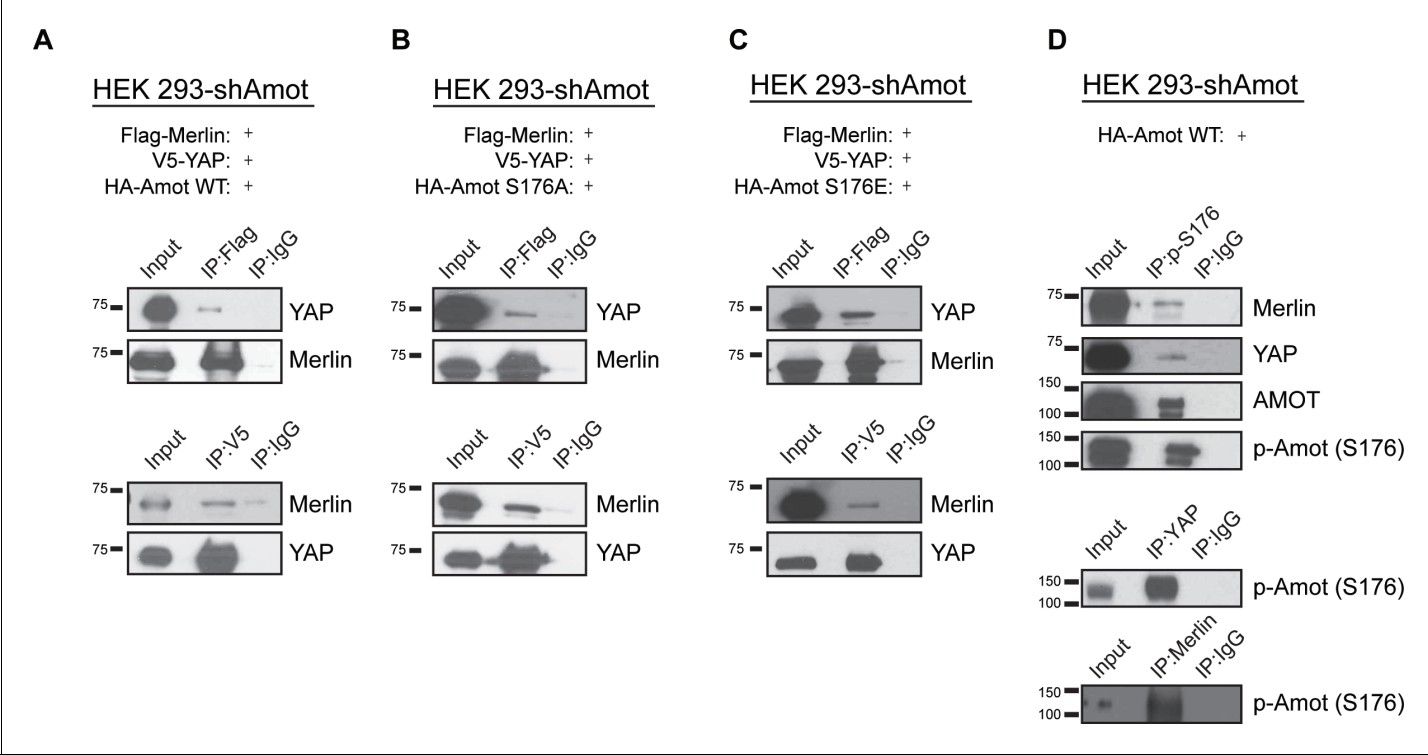

**Figure 3.** Phosphorylation status of Amot-p130^S176 does not impact formation of the Amot/YAP/Merlin complex. HEK293-shAmot cells were co-transfected with expression plasmids for Flag-Merlin, V5-YAP, and (**A**) HA-Amot-WT (**B**) HA-Amot-p130^S176A or (**C**) HA-Amot-p130^S176E. Total lysates (input) and Flag or V5 IPs were subjected to immunoblot analysis with anti-YAP and anti-Merlin antibodies as indicated. (**D**) HEK293-shAmot cells were co-transfected with an expression plasmid for Amot-p130. Total lysates (input) and IPs for phospho-Amot (Ser176), YAP, and Merlin were subjected to immunoblot analysis with indicated antibodies. The blots shown are representative of three independent biological replicates (n = 3).

The following figure supplements are available for figure 3:

**Figure supplement 1.** Analysis of exogenous Amot expression levels and distribution.

**Figure supplement 2.** Phosphorylation of Amot does not impact formation of YAP/Amot-p130 or Merlin/Amot-p130 complexes.

**Figure supplement 3.** Phosphorylation of Yap^S127 does not impact the formation of the Amot/YAP/Merlin complex.

Amot-p80 was unable to support the association between these proteins (*Figure 2C*). Moreover, mutation of the individual PPEY/LPTY motifs or combinations thereof, significantly impaired the association between Merlin and YAP (*Figure 2—figure supplement 2*). Taken together, these results show that Amot functions as a scaffold that supports the YAP-Merlin association.

## Angiomotin phosphorylation does not impair formation of a YAP/Merlin complex

Several reports have shown the importance of Amot post-translational modifications in regulating its function, specifically phosphorylation of Serine 176 (Serine 175 in humans) (*Adler et al., 2013a*; *Chan et al., 2013*; *Dai et al., 2013*; *Hirate et al., 2013*; *Paramasivam et al., 2011*). We therefore explored whether Amot phosphorylation regulates formation of the Amot/YAP/Merlin complex. HEK293-shAmot cells were transfected with expression vectors for Flag-Merlin, V5-YAP and either HA-Amot-p130, HA-Amot-p130^S176A (non-phosphorylated mutant) or HA-Amot-p130^S176E (phospho-mimetic mutant) and similar levels of the different Amot alleles were confirmed (*Figure 3—figure supplement 1*). IPs were carried out using anti-Flag or anti-V5 antibodies and the presence of Merlin and YAP was determined by western blotting. These analyses demonstrated that all three forms of Amot support the formation of the complex, as evidenced by the co-IP of Merlin and YAP

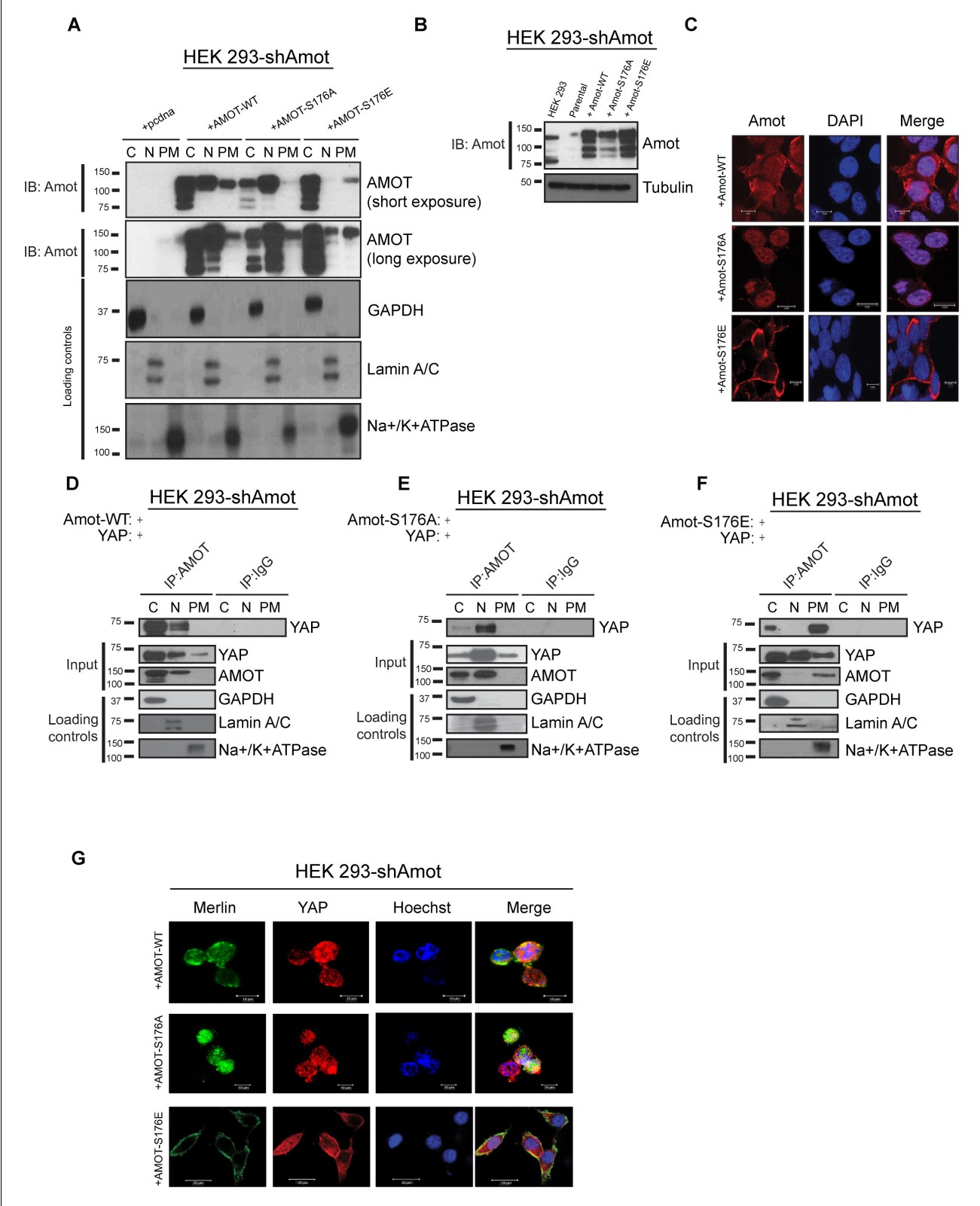

**Figure 4.** Amot-p130$^{S176}$ shifts localization of the YAP/Merlin complex. (**A**) Phosphorylation of Amot-p130 shifts its localization at the plasma membrane. HEK293-shAmot cells were transfected with Amot-WT, Amot-p130$^{S176A}$ or Amot-p130$^{S176E}$ expression plasmids and fractionated into cytoplasm (C), nuclear (N) and plasma membrane (PM) fractions. Immunoblot analysis was conducted using an anti-Amot antibody. GAPDH, Lamin A/C, and Na$^+$/K$^+$ATPase were using as controls for the cytoplasmic, nuclear, and plasma membrane fractions, respectively. (**B**) IB analysis was used to verify

*Figure 4 continued on next page*

*Figure 4 continued*

the transfection efficiency of the indicated constructs in total lysates of HEK293-shAmot cells. Tubulin was used as loading control. Blots shown are representative of three independent biological experiments (n = 3). (C) HEK293-shAmot cells were transfected with Amot-WT, Amot-p130$^{S176A}$ or Amot-p130$^{S176E}$ and subjected to immunofluorescence staining using an antibody against Amot. DAPI was used for nuclei fluorescence staining. Pictures show fields at 63x magnification and are representative of three independent biological replicates, in each of which 20 independent fields were examined. Scale bar = 10 µm. (D–F) Phosphorylation state of Amot-p130 mediates YAP localization. HEK293-shAmot cells were co-transfected with YAP and (D) Amot-WT, (E) Amot-p130$^{S176A}$ or (F) Amot-p130$^{S176E}$ and subjected to subcellular fractionation as in (A). Cell lysates (input) and Amot IPs of each one of the subcellular fractions were subjected to immunoblot analysis with anti-YAP antibodies as indicated. Loading controls were as in (A). All western blots shown are representative of three independent biological replicates (n = 3). (G) Double-immunofluorescence staining with anti-YAP and anti-Merlin antibodies on HEK293-shAmot cells transfected with expression vectors for Amot-WT, Amot-p130$^{S176A}$ or Amot-p130$^{S176E}$. Hoechst was used for staining of nuclei. Pictures show fields at 63x magnification and are representative of three independent biological replicates, in each of which 20 independent fields were examined. Scale bar = 10 µm.

The following figure supplements are available for figure 4:

**Figure supplement 1.** Amot phosphorylation regulates the localization of the Amot/YAP complex in human Schwann cells.

**Figure supplement 2.** Amot phosphorylation regulates the localization of the Amot/YAP complex in human hepatocellular carcinoma cells.

(*Figure 3A–C*). Further validation of these results were demonstrated by co-IP experiments in which Amot was IPed using an anti-HA antibody (*Figure 3—figure supplement 2*). In addition, we examined whether phosphorylated Amot is still able to bind both YAP and Merlin, by using an antibody that specifically binds phosphorylated S176 (p-S176). The antibody was used to IP phosphorylated Amot and co-IP of Merlin and YAP was assessed by western blotting. These experiments showed that Amot phosphorylated at S176 binds to both YAP and Merlin. Similarly, IP with antibodies against YAP or Merlin also showed co-IP with phosphorylated Amot (*Figure 3D*). Collectively, our findings demonstrate that Amot S176 phosphorylation state does not affect formation of the Amot/YAP/Merlin complex.

A major regulatory mechanism of YAP localization is through Lats1/2 phosphorylation of Ser127 promoting YAP cytoplasmic sequestration through 14-3-3 binding and concomitant sequestration from the nucleus (*Zhao et al., 2010b*, *2007*). We therefore evaluated whether YAP phosphorylation at Serine 127 plays a role in the establishment of the Amot/YAP/Merlin complex. Using similar IP approaches described above, we found that both phosphorylated YAP and constitutively active YAP$^{S127A}$ mutant retain their ability to associate with Amot/Merlin (*Figure 3—figure supplement 2*). Surprisingly, a YAP mutant where five major phosphorylation sites have been abolished (YAP-5A [*Zhao et al., 2010b*]) was no longer associated with the complex (*Figure 3—figure supplement 2C*).

## Amot$^{S176}$ phosphorylation regulates YAP localization at the plasma membrane

To determine whether phosphorylation induces a change in the sub-cellular localization of Amot/YAP/Merlin complex, 293-shAmot cells were transfected with expression vectors for Amot-p130, Amot-p130$^{S176A}$ or Amot-p130$^{S176E}$. The transfected cells were fractionated and the presence of Amot determined by Western blot analysis. In the cells expressing Amot-p130, the majority of this protein localized to cytoplasmic and nuclear fractions, and to a lesser extent in the plasma membrane fraction. Interestingly, Amot-p130$^{S176A}$ showed a marked increase in nuclear localization and almost no presence at the plasma membrane. In contrast, Amot-p130$^{S176E}$ showed a clear shift in localization towards the cytoplasm and plasma membrane (*Figure 4A*). Immunoblotting of total cell extracts confirmed similar expression levels of Amot-p130, Amot-p130$^{S176E}$ and Amot-p130$^{S176A}$ (*Figure 4B*). To determine the broader relevance of our findings in HEK293 cells to additional biological systems, we carried out a similar analysis in human hSC2λ Schwann cells and HepG2 hepatocellular carcinoma cells. Fractionation studies from these additional cell types followed a similar distribution pattern to the transfected 293-shAmot cells (*Figure 4—figure supplements 1A* and *2A*).

To complement the biochemical fractionation studies, we assessed localization of the different Amot proteins by IF staining in the 293-shAmot cells. These analyses confirmed the fractionation

findings, demonstrating a stronger intensity of Amot in the cytoplasm and nucleus of cells expressing Amot-p130, a shift to nuclear staining in Amot-p130$^{S176A}$ expressing cells and cytoplasmic/plasma membrane localization in Amot-p130$^{S176E}$ expressing cells (*Figure 4C*).

We next examined if the status of S176 can regulate localization of the Amot/YAP/Merlin complex. First, we examined Amot/YAP localization using co-IP studies in fractions prepared from 293-shAmot cells transfected with expression vectors for Amot-p130, Amot-p130$^{S176A}$ or Amot-p130$^{S176E}$. In agreement with our previous results, in 293-shAmot cells transfected with Amot-p130 and YAP, Amot co-IPed with YAP mainly in the cytoplasm but also in the nucleus with little to no interaction detected at the plasma membrane (*Figure 4D*). Importantly, in cells expressing the Amot-p130$^{S176A}$ allele, the Amot-YAP complex mainly localized to the nucleus with little to no interaction detected in the cytoplasm and plasma membrane (*Figure 4E*). Significantly, in cells expressing the Amot-p130$^{S176E}$ allele an increase in the Amot-YAP association was detected at the plasma membrane, in addition to the cytoplasmic fraction. Little to no interaction was detected in the nucleus (*Figure 4F*). Thus, the distribution of Amot/YAP complex mimics the distribution observed for Amot (*Figure 4A*). In addition to the analysis in the 293-shAmot cells, a similar analysis carried out in the hSC2λ and HepG2 cells confirmed a similar distribution pattern (*Figure 4—figure supplements 1B–D* and *2B–D*).

These findings were extended and confirmed by IF analysis of the YAP-Merlin association in 293-shAmot cells, where the localization of YAP and Merlin to the plasma membrane is significantly enhanced in cells expressing the Amot-p130$^{S176E}$ allele (*Figure 4G*). Overall our findings demonstrate that while Amot$^{S176}$ phosphorylation does not impact the formation of the Amot/YAP/Merlin complex, it mediates the localization of the complex through phosphorylation of Serine 176 that leads to recruitment of the complex to the plasma membrane or a shift to a nuclear localization in the hypophosphorylated state.

## Phosphorylation of Amot promotes localization to junctional structures at the plasma membrane

Previous studies showed binding of Amot to the tight junction-associated proteins Pals1 and Patj at the apical membrane (*Ernkvist et al., 2009*; *Sugihara-Mizuno et al., 2007*; *Wells et al., 2006*; *Yi et al., 2011*). To further characterize the mechanisms by which phosphorylated Amot shifts from the cytoplasm and nucleus to the plasma membrane, we examined the interactions between Amot-p130, Amot-p130$^{S176A}$ and Amot-p130$^{S176E}$ with Patj, Pals1 and E-cadherin. Using co-IP, we found that Flag-tagged Patj strongly associates with Amot-p130$^{S176E}$, compared to weaker or null interaction with Amot-p130 and Amot-p130$^{S176A}$, respectively. In a reciprocal approach, IP of the different Amot forms demonstrated a stronger association between Patj and Amot-p130$^{S176E}$ (*Figure 5A–C*). Similar results were obtained when examining the association of Pals1 or E-cadherin with Amot. While we were consistently able to co-IP Pals1 or E-cadherin with Amot-p130$^{S176E}$, this association was greatly diminished with Amot-p130$^{S176A}$ or wild type Amot-p130 (*Figure 5D–E*). These findings suggest that localization of phosphorylated Amot-p130 to the cell membrane is mediated through interactions with components of junctional structures, which remain to be identified.

Previous reports concluded that phosphorylation of Amot at Serine 176 reduces the association between Amot and F-actin (*Chan et al., 2013*; *Mana-Capelli et al., 2014*; *Dai et al., 2013*). To assess this directly in cells, we employed a co-precipitation approach relying on the high selectivity and binding affinity of phalloidin to F-actin, previously employed to identify F-actin binding proteins in cells (*Clarke and Mearow, 2013*; *Fulga et al., 2007*). Briefly, biotinylated phalloidin was used to specifically pull-down F-actin from 293-shAmot cells transfected with expression vectors for Amot-p130, Amot-p130$^{S176A}$ or Amot-p130$^{S176E}$ and levels of precipitated proteins were examined by western blotting. As shown in *Figure 5F*, similar amounts of all Amot proteins were present in the pull downs (Upper panel). Importantly, similar levels of Actin were present in all pull downs indicating the expression of the different Amot alleles did not affect cellular levels of F-actin (Lower panel).

## Amot phosphorylation regulates YAP-driven cellular proliferation and transcriptional activities

Given the impact of Amot-p130$^{S176}$ phosphorylation on localization of the Amot/YAP/Merlin complex, we conducted loss- and gain-of-function studies to gain insight into the functional significance

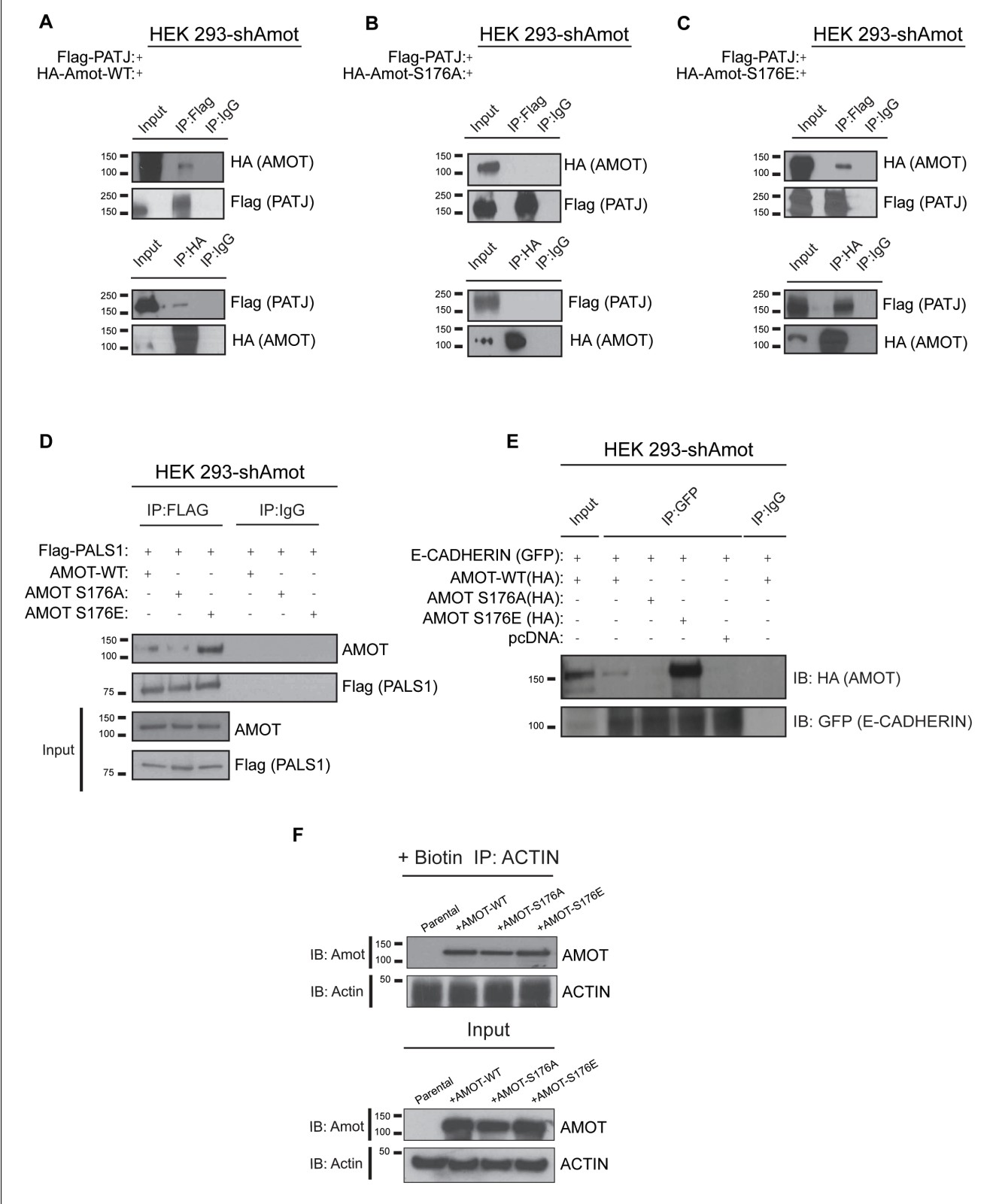

**Figure 5.** Amot[S176] status impacts binding to the junctional proteins PATJ, Pals1 and E-cadherin. HEK293-shAmot cells were co-transfected with Flag-PATJ and (**A**) HA-Amot-WT or (**B**) HA-Amot-p130[S176A] or (**C**) HA-Amot-p130[S176E]. Total lysates (Input) and Flag or HA IPs were subjected to immunoblot analysis with anti-HA and anti-Flag antibodies, as indicated. (**D–E**) HEK293-shAmot cells were co-transfected with (**D**) Flag-Pals1 or (**E**) E-cadherin and relevant Amot alleles as in panels (**A–C**). Total lysates (input) and Flag IP were subjected to IB analysis with anti-Amot and anti-Flag antibodies as

*Figure 5 continued on next page*

Figure 5 continued

indicated. (F) HEK293-shAmot cells were transfected with expression vectors for HA-Amot-WT, HA-Amot-p130$^{S176A}$ or HA-Amot-p130$^{S176E}$. Cells were extracted in F-actin stabilizing buffer and F-actin was pulled down using Biotinylated-phalloidin and streptavidin-coupled magnetic beads. Pull downs were then alayzed by western blotting using anti-Amot or Actin antibodies. The blots shown are representative of three biological replicates (n = 3).

of this phosphorylation. First, we determined whether Amot$^{S176}$ phosphorylation modulates cell proliferation. The 293-shAmot cells transfected with expression plasmids for Amot-p130 and Amot-p130$^{S176A}$ proliferated at increased rates compared to control transfected cells, as determined by cell counting and BrdU incorporation over a 96 hr period. In contrast, the Amot-p130$^{S176E}$ expressing cells exhibited a growth rate comparable to that of the control cells (*Figure 6A–B*). Similar results were obtained in hSC2λ and HepG2 cells transfected with the different Amot alleles (*Figure 6—figure supplement 1A–B*). These findings indicate that expression of Amot can promote the proliferation of these three different cell types and that this activity is diminished when Amot is phosphorylated at Serine 176.

As work from our group and others has shown that Angiomotin functions as a regulator of small G-proteins from the Rac1/cdc42 family (*Wells et al., 2006*; *Yi et al., 2011*), we assessed whether the increased proliferation observed in the Amot-p130 and Amot-p130$^{S176A}$ expressing cells is mediated by activation of Rac1. Towards this goal we analyzed the status of active Rac1 (Rac1-GTP) in these cells compared to control 293-shAmot cells. As expected, transfection with the different Amot alleles resulted in increased Rac1-GTP levels. However, we found no significant differences in Rac1-GTP levels between the cells, suggesting the status of Amot serine 176 phosphorylation does not affect Rac1 activation (*Figure 6C*).

We next determined whether the observed increased rates of cell proliferations induced by Amot-p130 and Amot-p130$^{S176A}$ expression are YAP-dependent, by introducing two independent siRNAs against YAP into these cells. Indeed, the increased proliferation afforded by Amot-p130 and Amot-p130$^{S176A}$ was completely inhibited by YAP knockdown (*Figure 6B* and *Figure 6—figure supplement 2A*). The expression of Amot-p130 and Amot-p130$^{S176A}$ and reduction of YAP levels was confirmed by immunoblotting (*Figure 6B*).

Next, we determined if Amot Serine 176 phosphorylation regulates YAP-mediated transcriptional activation. We used two independent luciferase reporter assays, the HIP/HOP-flash and 8xGTIIC luciferase reporters (*Leask and Abraham, 2006*; *Zhao et al., 2008*). The HIP (Hippo-YAP signaling incompetent promoter)/HOP (Hippo-YAP signaling optimal promoter)-flash reporters contain, respectively, seven mutated TEAD-binding sites and multimerized (x8) TEAD-binding sites from the promoter of YAP's direct target *CTGF* (*Leask and Abraham, 2006*; *Zhao et al., 2008*). The former is a negative control for HOP-flash activity (*Kim and Gumbiner, 2015*). In agreement with our previously reported findings (*Yi et al., 2013*), the activity of YAP in the HOP-reporter assay was suppressed in 293-shAmot cells, while the reintroduction of Amot fully rescued YAP activity (*Figure 6D–E*). Moreover, the expression of Amot-p130$^{S176A}$ resulted in a substantial increase in the activation of the HOP-flash reporter when compared to Amot. Significantly, Amot-p130$^{S176E}$ was unable to rescue the function of YAP in the 293-shAmot cells (*Figure 6D–E*). As expected, no significant changes in activation of the HIP reporter were observed under similar experimental conditions (*Figure 6D*). These observations were further confirmed using the GTIIC luciferase system, which has eight copies of TEAD-binding sequence driving the expression of the luciferase gene (*Davidson et al., 1988*; *Ota and Sasaki, 2008*; *Yi et al., 2013*) (*Figure 6—figure supplement 2B*).

To further evaluate the regulation of YAP's activity by Amot, we investigated the impact of Amot-p130, Amot-p130$^{S176A}$ and Amot-p130$^{S176E}$ expression in the 293-shAmot cells on known gene targets of YAP by quantitative real-time PCR. We focused on *Areg* (amphiregulin) and *ApoE* (Apolipoprotein E) as previous findings suggest Amot plays a role in their regulation (*Yi et al., 2013*). While expression of both Amot-p130 and Amot-p130$^{S176E}$ had only minor effects on levels of *Areg* and *ApoE*, the expression of Amot-p130$^{S176A}$ significantly induced up-regulation of *Areg* and *ApoE* by 2.5, and 3-fold respectively, relative to control cells (*Figure 6F*). Similar results were obtained in hSC2λ and HepG2 cells transfected with the different Amot alleles (*Figure 6—figure supplement 3*).

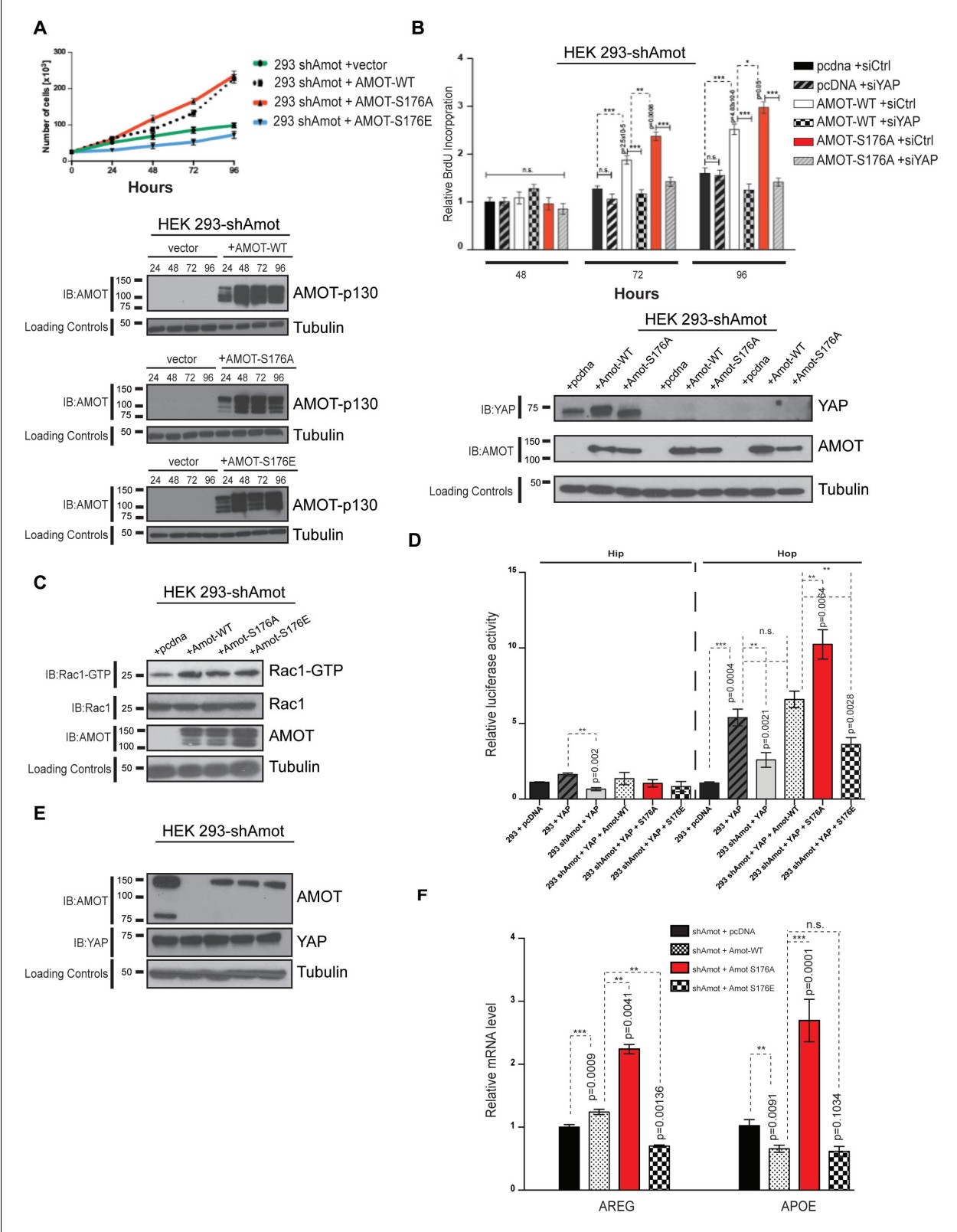

**Figure 6.** Amot[S176A] promotes the proliferative and transcriptional activities of YAP. (**A**) Amot[S176] regulates cellular proliferation. HEK293-shAmot cells were transiently transfected with indicated expression plasmids and total cell numbers were counted over 4 days. Means of each data point were calculated from three independent biological replicates conducted in triplicate. Error bars represent ±S.D. Immunoblot analysis was used to verify the transfection efficiency of the indicated Amot-p130 constructs. Tubulin was used as a loading control. The blots shown are representative of three

*Figure 6 continued on next page*

*Figure 6 continued*

biological replicates. (**B**) Amot expression drives proliferative phenotype that is YAP-dependent. HEK293-shAmot cells were co-transfected with the indicated expression plasmid and either a SMARTpool of siRNAs targeting YAP or a non-targeting control (siCtr). Levels of BrdU incorporation compared to HEK293-shAmot+pcDNA+siCtr (set to 1) were determined 48 hr, 72 hr, and 96 hr post co-transfection for all the conditions. Means were calculated from three biological replicates conducted in triplicate. Error bars represent ±S.D. Individual pairwise comparisons were assessed by Student's t-test, *p<0.05; **p<0.01; ***p<0.001; n.s. – non-significant. Exact p-values are indicated in the figure. Immunoblot analysis to confirm efficient knockdown of YAP using two independent siRNAs (siYAP-A and siYAP-B) (see Figure S6) and efficient overexpression of the indicated Amot-p130 constructs in HEK293-shAmot cells. Tubulin was used as a loading control. The blots shown are representative of three biological replicates (n = 3). (**C**) Amot-p130 serine 176 does not affect Rac1 activation. IB analysis of cell lysates from HEK293-shAmot cells expressing Amot-WT, Amot-p130$^{S176A}$ or Amot-p130$^{S176E}$ with anti-Rac1-GTP, anti-Rac1 and anti-Amot antibodies as indicated. Tubulin was used as a loading control. Cells were serum starved overnight and stimulated with 10 ng/mL EGF for 5'. The blots shown are representative of three biological replicates (n = 3). (**D**) Amot$^{S176}$ status regulates YAP transcriptional activity. HEK293 and HEK293-shAmot cells were transfected with indicated constructs and HIP-flash or HOP-flash reporters. Reporter's firefly luciferase activity was normalized to the levels of *Renilla* luciferase used as an internal control. The means of luciferase activity were calculated from three biological replicates conducted in quadruplicate. Error bars represent ±S.D. Individual pairwise comparisons were assessed by Student's t-test, **p<0.01; ***p<0.001; n.s. – non-significant. Exact p-values are indicated in the figure. (**E**) Immunoblot analysis showing efficient transfection of Amot-p130, Amot-p130 mutants, and YAP in cell lysates used in (**D**). Tubulin was used as a loading control. The blots shown are representative of three biological replicates. (**F**) Amot$^{S176}$ status regulates expression of endogenous YAP targets. Expression of the YAP target genes *Areg* and *ApoE* was probed in HEK293-shAmot cells expressing Amot-WT, Amot-p130$^{S176A}$ or Amot-p130$^{S176E}$ by quantitative real-time PCR. mRNA levels were compared with the empty vector control (set to 1). Means were calculated from *Ct* values in three independent biological replicates conducted in triplicate. GAPDH was used to normalize for variances in input cDNA. See *Table 1*. Error bars represent ±S.D. Individual pairwise comparisons were assessed by Student's t-test, **p<0.01; ***p<0.001; n.s. – non-significant. Exact p-values are indicated in the figure.

The following source data and figure supplements are available for figure 6:

**Source data 1.** Cell counts for HEK293 cells, treated as described *Figure 6A*.

**Figure supplement 1.** Amot$^{S176A}$ promotes proliferation of human Schwann and hepatocellular carcinoma cells.

**Figure supplement 1—source data 1.** Cell counts for hSCλ cells, treated as described in *Figure 6—figure supplement 1*.

**Figure supplement 1—source data 2.** Cell counts for hSCλ cells, treated as described *Figure 6—figure supplement 1*.

**Figure supplement 2.** Amot-p130$^{S176}$ pro-proliferative phenotype is YAP dependent and regulates YAP transcriptional activity.

**Figure supplement 2—source data 1.** Counts for BrdU incoporation.

**Figure supplement 2—source data 2.** Counts for luciferase activity.

**Figure supplement 3.** Amot-p130$^{S176}$ regulates YAP transcriptional activity.

**Figure supplement 3—source data 1.** Source data for qPCR analysis of AREG expression in hSC-lambda cells.

**Figure supplement 3—source data 2.** Source data for qPCR analysis of APOE expression in hSC-lambda cells.

**Figure supplement 3—source data 3.** Source data for qPCR analysis of AREG expression in HepG2 cells.

**Figure supplement 3—source data 4.** Source data for qPCR analysis of APOE expression in HepG2 cells.

Previously it was shown that Amot is part of transcriptionally active YAP-Tead-containing complex in the nuclei of adult liver and HEK293 cells (*Yi et al., 2013*). We hypothesized that Amot exerts its function in this complex in the dephosphorylated state. To test this, we performed reciprocal IP assays with antibodies against YAP or pan-Tead. We observed efficient co-IP of YAP and pan-Tead

**Table 1.** Primer sequences used in qPCR.

| Gene | Primers | |
|---|---|---|
| | Forward | Reverse |
| *Amot* | 5'-CAGCTTGCAGAGAAGGAATATGAG-3' | 5'-CTGGCTTTCTTTATTTTTTGCAAAG-3' |
| *ApoE* | 5'-AGGAACTGAGGGCGCTGA-3' | 5'-AGTTCCGATTTGTAGGCCTTCA-3' |
| *Areg* | 5'-TGATCCTCACAGCTGTTGCT-3' | 5'-TCCATTCTCTTGTCGAAGTTTCT-3' |
| *GAPDH* | 5' – GATCATCAGCAATGCCTCCT-3' | 5' – TGTGGTCATGAGTCCTTCCA-3' |

in the presence of Amot-p130 and Amot-p130$^{S176A}$ but not in the presence of Amot-p130$^{S176E}$ (*Figure 7A*). Moreover, we conducted Chromatin immunoprecipitation (ChIP) with Amot antibodies in 293-shAmot cells transfected with Amot-p130, Amot-p130$^{S176A}$ or an empty control vector, followed by RT-PCR analysis. These experiments showed a striking enrichment and binding of Amot-p130$^{S176A}$ to the promoter of *ApoE* and *Areg,* compared to Amot (*Figure 7B*). These findings suggest that phosphorylation of Amot serine 176 prevents YAP-mediated transcriptional activation by inhibiting Amot nuclear function as a co-factor in a YAP-Tead transcriptional complex (*Figure 7C*).

## Discussion

The role of Angiomotin regulating the Hippo/YAP pathway has so far been elusive, mainly due to conflicting reports suggesting that YAP regulation by Angiomotin is either positive or negative. To gain a deeper understanding of Amot function, we employed multiple cell types originating from tissues in which dysregulation of the Hippo-YAP pathway is observed under pathological conditions. Our study reveals that Angiomotin phosphorylation at serine 176 mediates YAP localization. This post-translational modification is the key event determining a promoting or repressing regulation of YAP by Angiomotin. We found that phosphorylation at serine 176 does not mediate Angiomotin's scaffolding functions but does impact the localization of a tertiary complex composed by Amot, YAP and Merlin, which is present in both the cytoplasm and nucleus for the wild type protein. The serine 176 phosphomimetic mutant (Amot$^{S176E}$) is recruited to the plasma membrane where it is found co-localized and associated with Patj, Pals1 and E-cadherin at junctional structures. Our previous studies indicate that Amot is required for YAP function in the nucleus (*Yi et al., 2013*). Thus, the relocation of Amot out of the nucleus and sequestration of Amot/YAP complex to the plasma membrane, function to prevent YAP from operating as a growth-promoting transcriptional activator. The non-phosphorylated mutant (Amot$^{S176A}$) is preferentially localized to the nucleus, where it facilitates YAP interactions with TEADs and concomitant activation of target gene transcription.

Angiomotin's involvement in regulation of Hippo-YAP signaling arises from studies showing that Amot functions as a scaffold protein for several components of this signaling cascade including YAP, Merlin, Kibra, Lats, and F-actin (*Moleirinho et al., 2014*). Our findings extend Amot scaffolding functions to the formation of the YAP-Merlin complex independent of Amot Ser176 phosphorylation, although it is possible that additional PTMs might regulate the complex specifically at different subcellular localizations. Multiple reports describe the role of Amot$^{S176}$ phosphorylation on Amot interactions with binding partners and function (*Adler et al., 2013b*; *Chan et al., 2013*; *Hirate et al., 2013*; *Mana-Capelli et al., 2014*). While these reports converge towards the idea that Amot serine 176 phosphorylation is mediated by Lats1/2 and that this impacts Amot's binding to F-actin, they differ significantly when examining the influence on Amot localization and function. Whilst the discussion of these differences merits a separate review, we highlight a few points that agree or conflict with our current findings. In regards to the effect of Amot phosphorylation on binding to F-actin or YAP, previous reports conclude that serine 176 phosphorylation impairs the interaction between Amot and F-actin and suggest that this can favor binding to YAP (*Chan et al., 2013*; *Mana-Capelli et al., 2014*; *Dai et al., 2013*). Our findings suggest that serine 176 phosphorylation does not impact association with F-actin. However, in agreement with one of these studies (*Dai et al., 2013*) our findings suggest that phosphorylation on Amot has little impact on the Amot-YAP association. Possible explanations for the differences between our current findings and previous reports could be attributed to the use of HEK293 cells expressing high levels of endogenous Amot, which

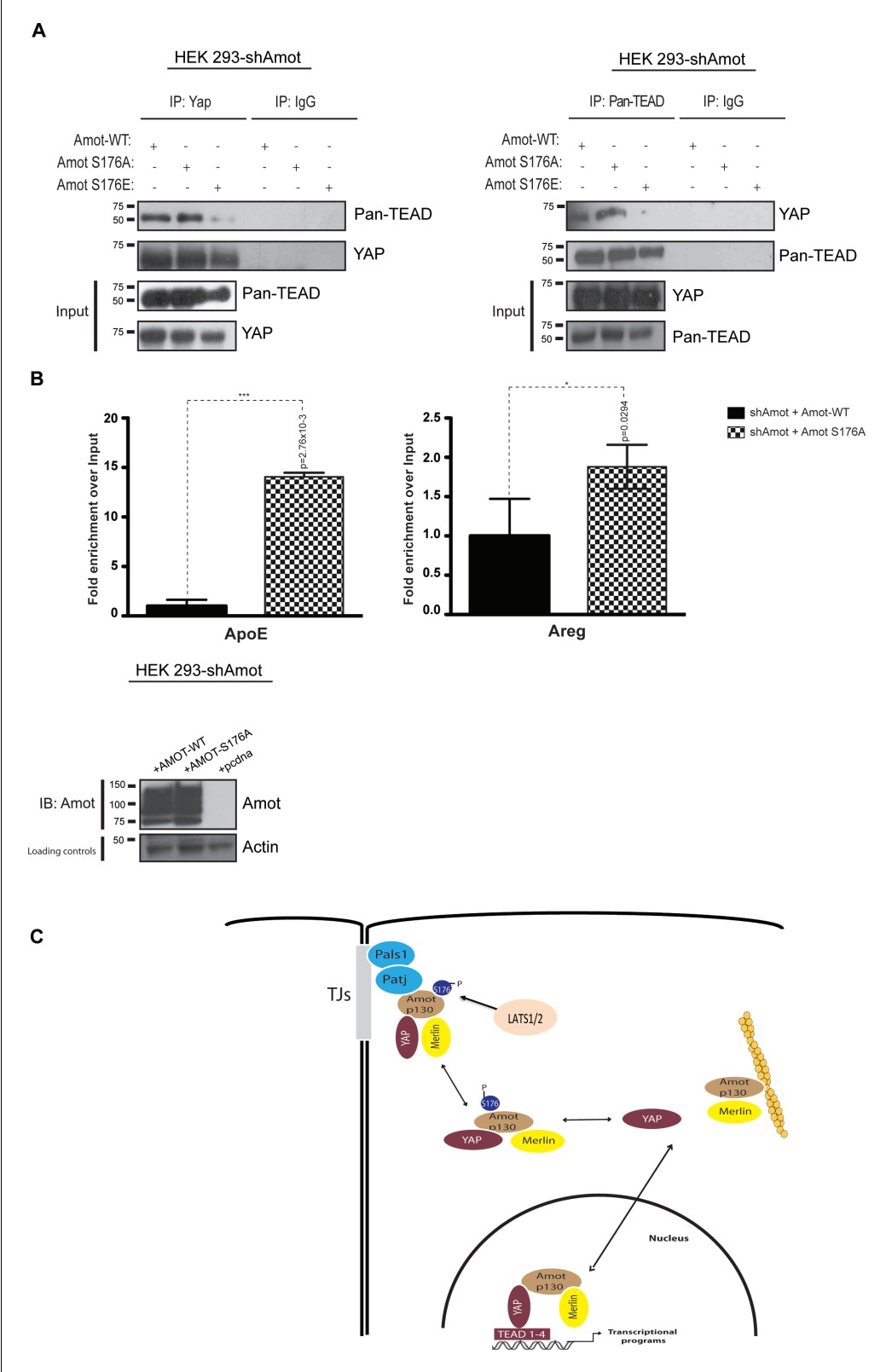

**Figure 7.** Amot[S176A] but not Amot[S176E] is required for formation of the nuclear Yap-Tead complex. HEK293-shAmot cells were co-transfected with Amot-WT or Amot-p130[S176A] or Amot-p130[S176E]. Total lysates (Input) and Pan-Tead or YAP IPs were subjected to immunoblot analysis with anti-Pan-Tead or anti-YAP antibodies, as indicated. The blots shown are representative of three independent biological replicates. (**B**) ChIP analysis of HEK293T-shAmot cells transfected with Amot-WT, Amot-p130[S176A] or an empty vector control. Real-time quantitative PCR was performed in eluted DNA using

*Figure 7 continued on next page*

*Figure 7 continued*

primers targeting the promoter regions of *ApoE* and *Areg*. See **Table 2**. The data show the means ±s.e.m. from three independent biological replicates (n = 3). Individual pairwise comparisons were assessed by Student's t-test, **p<0.01; ***p<0.001; n.s. – non-significant. Exact p-values are indicated in the figure. (C) Proposed model for YAP-Merlin complex regulation by Amot[S176]. Hypophosphorylation of Amot[S176] promotes translocation of the Amot-p130/YAP/Merlin complex from the cytoplasm to the nucleus where it binds to TEADs and activates YAP-dependent transcriptional programs. Conversely, phosphorylation of Amot[S176] induces cytoplasmic sequestration and plasma membrane localization of the tertiary complex. At the membrane, the complex associates with the junctional proteins Patj and Pals1 and YAP's nuclear functions are inhibited. S176 in blue indicates phosphorylation.

The following source data is available for figure 7:

**Source data 1.** Source data for qPCR analysis of ApoE expression in HEK293 cells.

**Source data 2.** Source data for qPCR analysis of AREG expression in HEK293 cells.

might mask some of the interactions with the mutated alleles. To circumvent this possibility, our studies incorporated HEK293 cells in which endogenous Amot expression was knocked-down and the different mutant alleles reintroduced. Additionally, previous studies used different cell types, which not only could impact the stability of the Amot/YAP interaction but also explain the observed differences in Amot localization. In MCF10A, MCF7 and MDA-MB-468 cells, hypophosphorylated Amot showed junctional localization and co-localization with F-actin (*Adler et al., 2013b*; *Chan et al., 2013*). Although we cannot exclude differences in the actin cytoskeleton between the different cell lines used in those studies, we observe enrichment of Amot[S176E] to the plasma membrane, in agreement with *Hirate et al. (2013)*, and enrichment of Amot[S176A] to the nucleus. Furthermore, our studies confirm previous findings and show that Amot is required for YAP activity and that the Amot[S176A] mutant displays prominent nuclear function by means of increased levels of YAP target gene expression, increased cell proliferation rates and higher colony formation capacity (*Adler et al., 2013b*; *Chan et al., 2013*; *Hirate et al., 2013*). Importantly, in vivo studies are needed to clarify the role of Amot in different cell and tissue types. We further extended the analysis of Amot phosphorylation on the Amot-YAP-Merlin complex, by determining whether its formation depends on YAP[S127] phosphorylation site. In agreement with previous reports showing that the Amot-YAP interaction is independent of phosphorylation at YAP[S127] (*Wang et al., 2011*; *Zhao et al., 2011*), we observed that the YAP-Merlin-Angiomotin complex also occurs independently of YAP[S127] (*Figure 3—figure supplement 3A and B*). Interestingly, mutation of serine in each of YAP's five HxRxxS consensus sites (S5A, *Zhao et al., 2010b*) precluded formation of the complex (*Figure 3—figure supplement 3C*). The molecular mechanisms underlying the reduced association between the S5A and the Amot-merlin complex remain to be elucidated. However, this is not likely as a result of increased nuclear localization of YAP-S5A, since wild-type Amot can also be found in the nucleus in complex with wild type YAP. Therefore, it is likely that at least one additional post-translational modification of YAP could regulate the formation of the Amot-YAP-Merlin complex and future studies are needed to identify this modification.

We found that Amot[S176] phosphorylation induces association of the YAP-Merlin complex with Patj, Pals1 and E-cadherin, and that plasma membrane sequestration inhibits YAP activity. Several reports have described the localization of Amot at junctions in epithelial and endothelial cells (*Bratt et al., 2005*; *Ernkvist et al., 2008*; *Patrie, 2005*; *Wells et al., 2006*) and its colocalization with Patj/Pals1 via its C-terminal PDZ-binding motifs (*Wells et al., 2006*). However, binding to Patj is not required for Amot's localization to the plasma membrane, as a mutant lacking the PDZ binding

**Table 2.** Primer sequences used in CHIP.

| Gene | Primers | |
| --- | --- | --- |
| | Forward | Reverse |
| *ApoE* | GCGTTCACTGTGGCCTGTCCA | GCATGGAGGACAGCCCTGGC |
| *Areg* | TGTTCTTCCCAGAAACCCTC | TTTACCTACACCATCTCACAGC |

motifs still localize to the cell cortex (*Sugihara-Mizuno et al., 2007*). This suggests that an additional mechanism can regulate Amot membrane localization, which is in agreement with our findings where phosphorylation of Amot$^{S176}$ induces localization of Amot-YAP-Merlin complex to the cell cortex. In HeLa, HEK293 and MDCK cells, Amot mediates YAP localization at the plasma membrane resulting in suppression of cell proliferation (*Chan et al., 2011*; *Paramasivam et al., 2011*; *Zhao et al., 2011*). Thus, we propose that phosphorylation of Amot triggers a shift of the YAP-Merlin complex from the nucleus to the plasma membrane, thus suppressing the nuclear activity of YAP and exerting cell growth and tumor suppressive functions (*Figure 7C*).

Amot associates with Merlin and Rich1 at junctional structures and inhibits Rac1 and downstream signaling into the MAPK pathway (*Yi et al., 2011*). Our results suggest that Amot$^{S176}$ phosphorylation does not impact the ability to modulate Rac1 activity at the cell membrane. Yet, it is possible that Amot$^{S176}$ phosphorylation modulates Merlin's nuclear function. Previous studies showed that nuclear accumulation of Merlin results in inhibition of the CRL4$^{DCAF1}$ E3 ubiquitin ligase. In *NF2*-mutant tumors, CRL4$^{DCAF1}$ ubiquitinates Lats, suppressing phosphorylation and activating YAP (*Li et al., 2014*, *2010*). In light of our findings, we can speculate that in the nucleus, Lats is inactivated by CRL4$^{DCAF1}$-driven ubiquitylation and becomes unable to phosphorylate Amot, resulting in an increase of YAP transcriptional activity. However, if Lats remains active and Amot phosphorylation occurs, Amot-YAP-Merlin tertiary complex translocates from the nucleus to the cytoplasm and plasma membrane, resulting in suppression of YAP nuclear functions. Future studies will be required to shed light on this possibility.

Another open question pertains to what mechanism/s regulate Angiomotin/YAP localization. Lats1/2 are obvious candidates, as they were shown to phosphorylate both YAP and Amot (*Adler et al., 2013b*; *Chan et al., 2013*; *Hirate et al., 2013*; *Mana-Capelli et al., 2014*). However, the role of Lats1/2 in regulation of these proteins and the relationship between Lats, YAP and Angiomotin is complex. For example, a number of reports suggest that Lats phosphorylates Amot, leading to reduced binding to F-actin and increased YAP binding and inhibition (*Chan et al., 2013*; *Mana-Capelli et al., 2014*). Moreover, Amot has been proposed to activate Lats2 and increase phosphorylation of YAP (*Paramasivam et al., 2011*; *Chan et al., 2013*). In contrast, we have previously shown that in multiple systems, Amot antagonizes the association of Lats and YAP and inhibits phosphorylation of YAP by Lats (*Yi et al., 2013*). Given the complexity of the interactions between Amot and YAP, addressing the role of Lats1/2 in regulating the translocation and activity of the Amot-YAP complex will require future studies. Another potential regulatory mechanism stems from the lack of interaction between Amot and YAP-S5A mentioned above. This finding suggests that additional phosphorylation events could regulate the complex. As a number of other kinases such as AMPK and CK1delta/epsilon have been shown to phosphorylate YAP (*Wang et al., 2015*; *Zhao et al., 2010a*). Further work will be required to determine the relevant phosphorylation sites and responsible kinases

In conclusion, our studies suggest a mechanism to explain the previous conflicting observations regarding the role of Amot by demonstrating that Amot$^{S176}$ phosphorylation state is a key event that dictates a positive or negative regulation of YAP by Amot by targeting the Amot-YAP-Merlin complex to the plasma membrane, sequestration in the cytoplasm or translocation to the nucleus.

## Materials and methods

### Plasmids and siRNAs

The expression plasmids for HA-Amot-p130, HA-Amot-p130$^{S176A}$ and HA-Amot-p130$^{S176E}$ were a gift from Dr. Hiroshi Sasaki (Kumamoto University, Japan). The following plasmids were previously described: psCMV-Pals1-Flag; PATJ-Flag (*Wells et al., 2006*; *Yi et al., 2011*); Flag or HA-tagged Merlin (*Kissil et al., 2002*, *2003*); pCMV-Flag-Amot-p80, pCMV-V5-YAP and pCMV-Amot-130 LPxY/PPxY mutants (*Yi et al., 2011*, *Yi et al., 2013*). The pcDNA4/His-MaxB-YAP-S127A (Plasmid #18988) and pCMV-Flag-YAP-5SA (Plasmid #27371) were from Addgene. Human Amot-p130 shRNA vectors have been previously described (*Yi et al., 2011*). siRNA duplexes targeting human YAP (ID #s20366) as well as non-targeting control siRNAs (ID #4390843) were from Thermo Fischer Scientific (Carlsbad, CA). The second siRNA targeting human YAP1 (5 FlexiTube #SI02662954) was purchased from Qiagen. siRNA targeting human angiomotin (ON-TARGETplus Human AMOT siRNA- smartpool

L-015417-01-0005) or control pool of siRNA (ON-TARGETplus Non-targeting pool-set of 4 LU-017595-01-0002).

## Cell culture, transfection, and infection conditions

HEK293 and HepG2 cells were purchased from the ATCC. hSC2λ cells were obtained from the laboratory of Dr. Margret Wallace (*Li et al., 2016*). All cell lines were authenticated by short tandem repeat (STR) DNA profiling (DDC Medical). Cells were tested every 3 months for mycoplasma contamination and confirmed free of contamination. Cells were maintained in low glucose Dulbecco's Modified Eagle's Medium (DMEM) (Gibco) supplemented with 10% fetal bovine serum (Atlas Biologicals) and antibiotics (100 units/ml penicillin and 100 μg/ml Streptomycin) (Gibco), at 37°C in a humidified atmosphere of 5% $CO_2$ (v/v). All experiments were carried out on cells grown to 70–80% confluency. Transfections were performed using Lipofectamine 2000 (Invitrogen, Carlsbad, CA) unless stated otherwise. Lentiviral infection of HEK293 cells was performed according to standard protocols. Briefly, HEK293T cells were co-transfected with packaging plasmids VSVG, Δ8.2, and either with pLKO.1-GFP or pLKO.1-shAmot constructs. Supernatants were collected 48 hr and 72 hr after transfection, and cells were infected with 4 mL of viral supernatant containing 4 mL of polybrene (8 μg/mL). After 48 hr, transduced cells were selected with puromycin (2 μg/mL) and this selection maintained for 72 hr.

## Antibodies

Rabbit polyclonal anti-Angiomotin was previously described (*Yi et al., 2011*) (IB: 1:1500). The following antibodies are available commercially: anti-phospho-Angiomotin (ABS1045 from EMD Millipore; 1:2000). Anti-HA tag (sc805, 1:1000); monoclonal anti-Merlin (E-2, 1:500); polyclonal anti-Merlin (C-18, 1:500); polyclonal anti-YAP (H-125, 1:1000); polyclonal anti-Lamin A/C (sc-6215 (N-18), 1:500); polyclonal anti-EGFR (sc03, 1:1000) were from Santa Cruz Biotechnologies. Anti-Flag tag (F1804, 1:1000); anti-GAPDH (68795, 1:10000); anti-tubulin (T5168, 1:1000); anti-actin (A4700; 1:10000) were from Sigma. Anti-V5 tag (ab27671, 1:2000) from Abcam. Anti-6x His tag (MA1-21315, 1:2000) from Thermo Scientific. Anti-phospho-YAP (4911, 1:1000); anti-pan-Tead (13295; 1:1000) from Cell Signaling Technologies. Anti-$Na^+/K^+$ATPase (a-5, 1:2500) from Developmental Studies Hybridoma Bank (University of Iowa). For immunofluorescence the following antibodies were used at the stated concentrations: rabbit monoclonal anti-YAP D8H1X (14074, 1:50), Cell Signaling Technologies; mouse monoclonal anti-Merlin (E-2; 1:50), Santa Cruz Biotechnologies; Rabbit polyclonal anti-Angiomotin (1: 200).

## Immunoprecipitation and immunoblotting

150 cm diameter dishes were transiently transfected with 10 μg of plasmid DNAs using Lipofectamine 2000 (Invitrogen). 48 hr later cell lysates were collected, washed twice in ice-cold PBS, and lysed with radioimmuno precipitation assay buffer (RIPA: 50 mM Tris-HCl; 150 mM NaCl; 1% NP40; 0.1% sodium dodecyl sulfate; 0.5% sodium deoxycholate; 1:25 protease inhibitor cocktail and phosphatase inhibitors (Roche Applied Science)). 1 mg of protein was immunoprecipitated with Protein A/G resins (#20333; #20398; Thermo Scientific) and indicated antibody with gentle rotation at 4°C, overnight. Immunoprecipitates were washed four times in RIPA buffer, and bound proteins were dissociated in 25 μL of 1x loading dye (25 mM Tris-HCl pH 6.8, 4% SDS, 5% glycerol, bromophenol blue). Eluted proteins were separated on SDS/10% polyacrylamide gel and transferred onto Immobilon-P membranes (Millipore). To prevent nonspecific binding, membranes were incubated in blocking buffer (5% skimmed dried milk, 33.3 mM Tris-HCl, 16.68 mM Tris base, 138 mM NaCl, 2.7 mM KCl, 0.1% Tween-20) with agitation for 1 hr at room temperature, followed by immediate incubation with specific antibodies diluted in either 5% BSA or blocking buffer, overnight. Membranes were then washed three times in washing buffer (33.3 mM Tris-HCl, 16.68 mM Tris base; 138 mM NaCl; 2.7 mM KCl; 0.1% Tween-20), incubated for 1 hr at room temperature with goat anti-mouse HRP-conjugated antibody (sc-2005; 1:10000) or Protein A-HRP linked (NA9120V; 1: 2000; GE Healthcare) and protein expression was detected by chemiluminescence using ECL (#RPN2106 or #RPN2236, GE Healthcare).

## F-actin pull down assay

Cells were collected and lysed with actin stabilization buffer (1% Triton-X 100, 0.1% SDS, 10 mM EDTA, 1% sodium deoxycholate, 200 pM sodium vanadate, 200 pM NaF, 1 complete protease inhibitor cocktail tablet, 0.5 mM ATP and Tris-Buffered Saline, pH 7.4). Biotinylated-phalloidin (5 µg, Thermo-Fisher Scientific) was added to samples and incubated for 60' at 4°C with constant rotation. Subsequently, 20 µL of streptavidin-coupled magnetic Dynabeads were added, incubated for 60' at 4°C with constant rotation. The magnetic beads were isolated with a magnet, washed 3X with PBS and analyzed as described.

## Immunofluorescence

Immunofluorescence staining was carried out as previously described (*Li et al., 2010*). Briefly, cells were grown on coverslips and transfected the next day using Lipofectamine. After 48 hr, coverslips were fixed with 4% paraformaldehyde in PBS for 20 min at room temperature and incubated in permeabilization buffer (0.3% sodium deoxycholate and 0.3% Triton-X in PBS) for 30 min on ice. Coverslips were then washed twice in PBS and blocked in 5% goat serum/0.3% Triton-X in PBS for 1 hr at room temperature followed by overnight incubation with indicated primary antibodies. After three PBS washes, coverslips were incubated with goat anti-mouse Alexa Fluor 488 (#A11029, 1:400; Life Technologies), goat anti-rabbit Alexa Fluor 568 (#A11011, 1:400; Life Technologies) or both secondary antibodies for 1 hr at room temperature. Cells were stained 5 min with 1 µg/mL of DAPI or Hoechst and mounted in Vectashield. Slides were examined using confocal microscopy (LSM 780; Carl Zeiss; Plan Neofluar 63x/1.3 NA Korr differential interference contrast M27 objective in water) at room temperature. Digitalized images were assembled using ZEN 2011 (64 bit) software (Carl Zeiss).

## Sub-cellular fractionation

Cells were incubated on ice for 20 min in ice-cold cytoplasmic buffer (20 mM Tris-HCl pH 7.4, 150 mM KCl, 1.5 mM MgCl$_2$, 1 mM PMSF, 1 mM DTT, 0.5%Nonidet P-40, protease inhibitor mixture), centrifuged at 4000 g for 5' at 4°C. Supernatant was kept for the cytosolic and plasma membrane fractions. The pellet was washed five times with nuclear washing buffer (10 mM HEPES pH7.9, 10 mM KCl, 1.5 mM MgCl$_2$, 0.34 M Sucrose, and complete protease inhibitor mixture), lysed on ice for 20 min in 400 µl of RIPA buffer and centrifuged at 13,000 g for 10 min at 4°C. Supernatant was kept as the nuclear fraction. For the cytosolic and plasma membrane fractions the supernatant was spun at 200,000 g for 30 min at 4°C and the resultant supernatant centrifuged again at 13,000g for 5 min at 4°C and kept as the cytosolic fraction. The pellet was washed in 500 µl lysis buffer (50 mM Tris pH 7.4, 1 mM EDTA, 2.5 mm MgCl$_2$, 150 mM NaCl, and complete protease inhibitor mixture) and re-sedimented at 200,000g for 30 min at 4°C. The pellet was then resuspended in ice-cold IP buffer (1 M Tris-HCl pH 7.4, 4 M NaCl, 10% (w/v) Triton X-100, and complete protease inhibitor mixture), incubated on a rocker for 30 min at 4°C, and cleared by centrifugation at 13,000g for 15 min at 4°C. The supernatant was kept as the plasma membrane fraction.

## Immunoprecipitation and Rac1-GTP pull-down

HEK293-shAmot cells were transfected with 8 µg of plasmid DNA (empty vector control/ wild type Amot-p130/Amot$^{S176A}$/Amot$^{S176E}$) and 48 hr later lysed in RIPA buffer and precipitated with indicated antibody overnight. Rac1-GTP was pulled down according to the manufacturer's instructions (Millipore, #17–441).

## BrdU incorporation assay

HEK293-shAmot cells were seeded at a cell density of $2 \times 10^5$ cells per well in 6-well plates. On the following day cells were co-transfected with 40 nM of siRNAs (targeting either YAP or a non-targeting siRNA duplex (control)), and 2 µg of plasmid DNA (empty vector control/ wild type Amot-p130/Amot$^{S176A}$/Amot$^{S176E}$) using TransIT-LT1 and TransIT-TKO (Mirus, Fisher Scientific, Illinois) according to manufacturer's instructions. After 24 hr, cells were trypsinized and $2 \times 10^4$ cells/well were reseeded in sterile 96-well tissue culture plates. 48 hr, 72 hr, and 96 hr after co-transfection, plates were processed and analysed using BrdU cell proliferation assay kit following manufacturer's

instructions (Millipore, #2750). Plates were read using a spectrophotometer microplate reader set to 450 nm (SpectraMax M5; Molecular Devices).

## Luciferase assay

The HIP/HOP-flash luciferase reporter system was a kind gift from Dr. Barry Gumbiner (*Gumbiner and Kim, 2014*) and the GTIIC-luciferase reporter was previously described (*Yi et al., 2013*). HEK293 and HEK293-shAmot cells were seeded at a concentration of $40 \times 10^3$ cells/50 µl in 96-well plates. On the following day the indicated luciferase reporters and plasmids were transfected to a total of 160 ng and incubated overnight at 37°C. Luciferase activity was then measured with Dual Luciferase Reporter Assay System (Promega; #E1910) according to manufacturer's instructions. The reporter's firefly luciferase activity was normalized to the levels of *Renilla* luciferase used as an internal control reporter. The relative luciferase activity displayed on the Y-axis indicates the ratio between Firefly/*Renilla* luciferase activities.

## RNA extraction and quantitative real-time PCR (qPCR)

Extraction of RNA from cell lysates was performed using Qiagen RNeasy kit (Qiagen) followed by cDNA synthesis of 1 µg DNase-digested RNA, using SuperScript III First-Strand Synthesis System for quantitative RT–PCR (Invitrogen) according to manufacturer's instructions. Quantitative PCR of the synthesized cDNA was conducted using SYBR Green 2x Master Mix (Applied Biosystems) according to the manufacturer's protocol. 10 ng of each sample were used in each analysis. Real-time quantitative RT-PCR reactions were performed on StepOnePlus Real-Time PCR System (Applied Biosystems) and analysed using StepOne Software v2.2.2. All measurements were conducted three times in triplicate and standardized to the levels of GADPH. Relative changes in gene expression were calculated according to the $2^{-\Delta\Delta CT}$ algorithm (*Livak et al., 2001*). Sequence of the qPCR primers is provided in *Table 1*.

## Chromatin immunoprecipitation (ChIP) and qPCR analysis

20 million HEK293T-shAmot cells were fixed with 1% formaldehyde for 10 min at RT. Fixation was halted with 125 mM glycine for 5 min at RT. Fixed cells were washed twice with cold PBS. Cell pellets were then resuspended in ChIP lysis buffer and chromatin was sheared with sonicator to obtain 0.3–0.5 kb DNA fragments. Angiomotin antibody (5 µg) and Dynabeads Protein A were added to the cell lysate and incubated overnight at 4°C. Beads were washed with buffer 1 (150 mM NaCL, 20 mM TrisCl pH 8.0, 5 mM EDTA, 65% w/v sucrose, 10% Triton-X-100, 20% SDS) and then washed with TE buffer. DNA was eluted by resuspending the beads in TE/1%SDS. ChIP DNA and Input were treated with RNase A (5 µg) for 1 hr at 37°C. Proteinase K (0.5 mg/mL) was added and incubated overnight at 65°C to reverse crosslinking. DNA was then purified phenol:chloroform and resuspended in a 30 µL of elution buffer. DNA was used for real time-PCR using SYBR Green PCR kit. A standard dilution curve was obtained for each Input and 1 µL of ChIP DNA was used in each PCR reaction. Melt curves were analyzed to confirm specificity of the amplified target.

## Acknowledgements

We thank Drs. Margaret Wallace and Hua Li (University of Florida) for the immortalized human Schwann cells; Dr. Hiroshi Sasaki (Institute of Molecular Embryology and Genetics, Japan) for providing the HA-Amotp130/S176A/S176E/pcDNA3.1-pA83 plasmids and Dr. Barry M Gumbiner (University of Virginia Health Sciences Center) for HIP/HOP-flash luciferase reporters. This work was supported by the NIH (NS077952 and CA124495 to JK). SM is a recipient of the Young Investigator Award from Children's Tumor Foundation.

## Additional information

### Funding

| Funder | Grant reference number | Author |
| --- | --- | --- |
| National Cancer Institute | CA124495 | Joseph L Kissil |

| National Institute of Neurological Disorders and Stroke | NS077952 | Joseph L Kissil |
| Children's Tumor Foundation | YIA | Susana Moleirinho |

The funders had no role in study design, data collection and interpretation, or the decision to submit the work for publication.

## Author contributions

SM, Conceptualization, Formal analysis, Investigation, Methodology, Writing—original draft, Writing—review and editing; SH, Formal analysis, Investigation, Methodology; VM, GC, Formal analysis, Investigation; ST, UE, Investigation, Methodology; JLK, Conceptualization, Formal analysis, Supervision, Funding acquisition, Methodology, Writing—original draft, Project administration, Writing—review and editing

## Author ORCIDs

Joseph L Kissil, http://orcid.org/0000-0002-6471-346X

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
