## [Decision Letter]

Thank you for submitting your article "Regulation of localization and function of the transcriptional co-activator YAP by angiomotin" for consideration by *eLife*. Your article has been favorably evaluated by Anna Akhmanova (Senior Editor) and three reviewers, one of whom is a member of our Board of Reviewing Editors. The reviewers have opted to remain anonymous.

The reviewers have discussed the reviews with one another and the Reviewing Editor has drafted this decision to help you prepare a revised submission.

In this manuscript, the authors demonstrate that Amot acts as a scaffold to recruit YAP to Merlin, and that the formation of this complex is not affected by the phosphorylation of Amot at S176. Interestingly, however, the mutation of S176 to E induces a clear relocalization of Amot from the nucleus to the plasma membrane and the cytosolic fraction (with the plasma membrane localization very apparent by microscopy), while mutation to A result in strong nuclear signal. This is accompanied by a redistribution of YAP and Merlin (and in consequence of the complex) to the plasma membrane in the S176E cells, perhaps through augmented association with Pals1/Patj. The consequence of the phosphorylation status of Amot on proliferation / YAP activity was investigated, revealing strong activity of the S to A mutant in all assays, and inactivity of the S176E mutant. It provides evidence that AMOT facilitates YAP nuclear and transcriptional function, unless phosphorylated at ser 176, which facilitates AMOT interaction with plasma membrane proteins. The authors’ findings disagree with some previous conclusions about the role of AMOT, which they address well, and which makes the finding of importance.

All reviewers thought that this work was novel and interesting. However additional controls are considered necessary to demonstrate some of the reagents are specific, quantification is needed, and some of the conclusions should be moderated or caveats considered in the Discussion.

1) What is the expression level of the ectopically expressed Amot, compared to that endogenously expressed in HEK cells? Almost the entire analysis depends on exogenous expression of AMOT and AMOT mutants. It is well known that overexpression of proteins often leads to their accumulation in inappropriate compartments. Since so much of the conclusions and model depend on the idea of compartments and pools, it is essential that the authors ensure that their proteins are expressed at normal cellular levels, and that all mutants are expressed at similar levels

2) Ectopically expressed PATJ and PALS1 co-Ips with Amot and AmotS>E, but not AmotS>A. This may be due to Amot localization at the membrane versus nucleus. Is association of Amot with Patj/PALS1 really needed for the membrane localization of the complex? The authors should knockdown Patj/PALS1 (to see if localization of AmotS>E changes) to support this conclusion.

3) Could the interaction (or non-interaction) of the Amot constructs with Patj and Pals1 be only a reflection of the differential proportion of the Amot constructs in the plasma membrane fraction where Patj and Pals1 reside? (i.e. is the differential association the cause of the localization or the consequence of altered localization?). As presented, this set of experiments is supportive of the differential localization, but does not necessarily demonstrate a direct role for association with Patj/Pals1 (especially as the PDZ motif is dispensable for Amot localization to the cortex). Is the association with Patj/Pals1 bringing Amot to the cortex? If more direct evidence for a preferential association of Amot S176E to Patj/Pals1 cannot be experimentally determined, the authors may want to revise the presentation of the associated results and conclusions.

Specific comments/suggestions to address in the Discussion:

4) Is it possible that overexpressed protein does not reveal a change in affinity upon altered phosphorylation?

5) The authors should discuss the 5A YAP results. Do they believe this can be explained by increased nuclear localization of 5AYAP?

6) The authors ignore the role of Lats, which is also thought to bind to AMOT, and the role of its phosphorylation of YAP in YAP regulation. Most studies on the hippo pathway agree that this phosphorylation event controls both the levels and nuclear-cytoplasmic distribution of YAP, but these two properties are not addressed. The authors do show that phosphorylation of YAP on ser 127 does not affect binding to AMOT, but do not show how AMOT regulates YAP phosphorylation or stability. Also, they do in fact see a change in binding of 5 ser > ala YAP mutations, which have been shown to regulate YAP function in addition to s127. This difference must be considered.

7) There is no consideration of any upstream regulation of the pathway in either the experimental analysis or in the evaluation of the models and literature. What controls AMOT phosphorylation? Are Mst/hippo kinases involved in AMOT regulation? Are the authors imagining that this regulation has something to do with cell-cell contact control of the pathway? Or control by soluble growth factors and signaling receptors? In the absence of this broader context, it is very difficult to think about mechanisms and models. At the least this should be touched on in the Discussion.

8) Their model implies a constitutive binding of YAP to AMOT in the control YAP localization by AMOT, without considering catalytic effects, especially those that might be conferred by Lats phosphorylation. Similar to other models, it invokes simple sequestration at the membrane via AMOT-PATJ binding to restrain YAP. Yet only a small fraction of S176 AMOT seems to associate with membranes or PATJ. Similarly, the model implies that the cytoplasmic and nuclear pools of YAP are all bound to AMOT. If a large fraction of YAP in the cytosol or nucleus is free, different mechanisms from the one proposed must be involved in controlling its localization.

9) The authors discuss previous studies showing that AMOT interacts with the actin cytoskeleton and that this interaction is regulated by AMOT phosphorylation. However, they do not address this potentially important pool of AMOT in their study. How does the actin associated pool fractionate in their protocol – with cytoplasmic, nuclear, or membrane fractions?

10) Figure 5 vs. A and C. The amount of flag PATJ IPed by anti-flag seems extremely low in this panel compared to others, so lack of HA-AMOT S176A co-IPing doesn't seem to be a well-controlled finding.

11) Others have reported that cadherins can also interact with merlin-AMOT complexes at the membrane in addition to PATJ. Do the authors think that this may also be a significant interaction worth examining?

12) The interpretation that the interaction between YAP and Merlin is predominantly occurring in the nucleus and the cytoplasm is supported by the data, but quantification of these experiments would help providing an assessment as to whether this simply reflect the relative levels of YAP across the different fractions or whether there is evidence for a truly differential interaction across fractions. [This would also be consistent with all the results presented for the phosphorylation experiments, so I would suggest considering this possibility in the text]. Perhaps an alternative way to provide support for a compartment-specific interaction would be to employ a fluorescence-based approach (e.g. split GFP, FRET).

13) The presentation of the western blots (IP/western) should be accompanied by quantification. I was surprised that experiments that I would assume should be loaded on the same SDS page gel (e.g. different constructs for the rescue experiments in Figure 3) were shown as separate panels. In the absence of quantification, this makes it very difficult to comment on whether a reduction/increase in complex formation occurs following mutation at this site (though clearly, the interaction is not abrogated by these mutations).

14) What is the intracellular localization of endogenous Merlin (and YAP)? The localization experiments currently presented all depend upon transient transfection.

15) For Figure 2, please provide supporting evidence for efficient Amot KD (if the same results have been also obtained with a different Amot hairpin, please also add the results in supplementary to confirm the specificity of the effect on complex formation).

16) If the phosphospecific antibody works in IF, it would be nice to provide additional evidence that the phosphorylated form is enriched in the plasma membrane fraction (alternatively, this could be done by immunoblotting). This would support the results obtained with the S to E mutation.

17) What is the relative level of expression (in relation to the endogenous) of the different Amot constructs? Similarly, the authors may want to comment on the expression levels of the other constructs they used in relation to the respective endogenous proteins. Could some of the results observed due to overexpression / tagging artifacts?

18) Concerning the Figure 3—figure supplement 2 (as per Discussion), since a YAP protein in which the 5 serines are mutated to alanines predominantly resides in the nucleus, could it be that the nuclear sequestration is the primary cause of the absence of complex formation?

19) Other comments: – this likely happened through PDF generation during submission, but the quality of the images is problematic. Labels do not display clearly, and microscopy images are extremely difficult to evaluate. The authors need to provide high quality images with the revision.

---

## [Author Response]

*[…] All reviewers thought that this work was novel and interesting. However additional controls are considered necessary to demonstrate some of the reagents are specific, quantification is needed, and some of the conclusions should be moderated or caveats considered in the Discussion.*

*1) What is the expression level of the ectopically expressed Amot, compared to that endogenously expressed in HEK cells? Almost the entire analysis depends on exogenous expression of AMOT and AMOT mutants. It is well known that overexpression of proteins often leads to their accumulation in inappropriate compartments. Since so much of the conclusions and model depend on the idea of compartments and pools, it is essential that the authors ensure that their proteins are expressed at normal cellular levels, and that all mutants are expressed at similar levels*

We agree with the reviewers’ comments. Assessment of expression of the different Amot alleles and comparison to endogenous Amot levels are now included in Figure 3—figure supplement 1. Overall, we find the levels of exogenous Amot to be approximately 1.5 to 2 –fold higher than endogenous. Importantly, this overexpression does not alter the localization of wild type Amot, as exogenous FLAG-tagged Angiomotin shows a similar subcellular distribution to the endogenous protein (Figure 3—figure supplement 1).

*2) Ectopically expressed PATJ and PALS1 co-Ips with Amot and AmotS>E, but not AmotS>A. This may be due to Amot localization at the membrane versus nucleus. Is association of Amot with Patj/PALS1 really needed for the membrane localization of the complex? The authors should knockdown Patj/PALS1 (to see if localization of AmotS>E changes) to support this conclusion.*

In spite our extensive attempts, using several different shRNA/siRNAs, we were unable to achieve efficient knockdown of PATJ/PALS1 and thus unable to assess whether the membrane localization of AmotS>E is changed upon loss of these proteins. In addition, while responding to other review comments, we are able to detect enhanced association of the AmotS>E mutant with E-cadherin (Figure 5). Therefore, while our results support a differential localization of this mutant to junctional structures, we cannot demonstrate a direct role for PATJ, PALS1 or E-cadherin in the localization of AmotS>E. We have revised the presentation of results and conclusions to reflect this.

*3) Could the interaction (or non-interaction) of the Amot constructs with Patj and Pals1 be only a reflection of the differential proportion of the Amot constructs in the plasma membrane fraction where Patj and Pals1 reside? (i.e. is the differential association the cause of the localization or the consequence of altered localization?). As presented, this set of experiments is supportive of the differential localization, but does not necessarily demonstrate a direct role for association with Patj/Pals1 (especially as the PDZ motif is dispensable for Amot localization to the cortex). Is the association with Patj/Pals1 bringing Amot to the cortex? If more direct evidence for a preferential association of Amot S176E to Patj/Pals1 cannot be experimentally determined, the authors may want to revise the presentation of the associated results and conclusions.*

Please see our response to question 2.

*Specific comments/suggestions to address in the Discussion:*

*4) Is it possible that overexpressed protein does not reveal a change in affinity upon altered phosphorylation?*

While this is a possibility, we believe the likelihood for this is low. If one hypothesizes that phosphorylation at S176 has an impact on the affinity of Amot towards merlin or YAP, it would be expected that the S176A or S176E mutants would have different effects on affinity and association with their binding partners, which is not what we observed in our studies.

*5) The authors should discuss the 5A YAP results. Do they believe this can be explained by increased nuclear localization of 5AYAP?*

The literature indicates the S5A mutant is preferentially localized to the nucleus. The experiment shown in Figure 3—figure supplement 2 suggests that the YAP-S5A mutant is no longer associated with the Amot-merlin complex. We don’t think it has to do with the increased nuclear localization of YAP-S5A, since wild-type Amot (which was used in the study shown in Figure 3—figure supplement 2) can also be found in the nucleus in complex with wild type YAP (see examples in Figure 4). Clearly additional studies are needed to elucidate the mechanisms involved. We included a discussion of this in the manuscript.

*6) The authors ignore the role of Lats, which is also thought to bind to AMOT, and the role of its phosphorylation of YAP in YAP regulation. Most studies on the hippo pathway agree that this phosphorylation event controls both the levels and nuclear-cytoplasmic distribution of YAP, but these two properties are not addressed. The authors do show that phosphorylation of YAP on ser 127 does not affect binding to AMOT, but do not show how AMOT regulates YAP phosphorylation or stability. Also, they do in fact see a change in binding of 5 ser > ala YAP mutations, which have been shown to regulate YAP function in addition to s127. This difference must be considered.*

Although we did briefly discuss the role of Lats in the paper, the role of Lats is complicated and not straightforward. Indeed Lats1/2 phosphorylate and regulate YAP localization and levels and a number of studies have shown Lats regulates phosphorylation of Amot on S176. However, the relationship between Lats, YAP and Angiomotin is complex. For example, a number of papers suggest that Lats phosphorylates Amot, leading to reduced binding to F-actin and increased YAP binding and inhibition (Chan, JBC, 2013 and Mana-Capelli, MBC, 2014). Moreover, Amot has been proposed to activate Lats2 and increase phosphorylation of YAP (Paramasivam, MBC, 2011 and Chan, JBC, 2013). In contrast, we have previously shown that in multiple systems, Amot antagonizes the association of Lats and YAP and inhibits phosphorylation of YAP by Lats (Yi, Sci Signal, 2013). Given the complexity of the interactions between YAP, Amot and YAP, we believe this question requires a more extensive and in depth analysis which is beyond the scope of the current manuscript. Some of the issues raised in our response are also addressed in the Discussion of the manuscript.

*7) There is no consideration of any upstream regulation of the pathway in either the experimental analysis or in the evaluation of the models and literature. What controls AMOT phosphorylation? Are Mst/hippo kinases involved in AMOT regulation? Are the authors imagining that this regulation has something to do with cell-cell contact control of the pathway? Or control by soluble growth factors and signaling receptors? In the absence of this broader context, it is very difficult to think about mechanisms and models. At the least this should be touched on in the Discussion.*

We agree with this point. We initially left this out due to space considerations. In particular, while are many possible scenarios involving regulation of Amot activity by upstream events, since S176 has been shown to be phosphorylated by Lats, we can speculate that many of the upstream events regulating the Hippo cascade will also regulate Amot. We now include this in the Discussion.

*8) Their model implies a constitutive binding of YAP to AMOT in the control YAP localization by AMOT, without considering catalytic effects, especially those that might be conferred by Lats phosphorylation. Similar to other models, it invokes simple sequestration at the membrane via AMOT-PATJ binding to restrain YAP. Yet only a small fraction of S176 AMOT seems to associate with membranes or PATJ. Similarly, the model implies that the cytoplasmic and nuclear pools of YAP are all bound to AMOT. If a large fraction of YAP in the cytosol or nucleus is free, different mechanisms from the one proposed must be involved in controlling its localization.*

We agree that examining the dynamics of the association of the complex would be useful, however, given space and time limitations we did not yet get to do these types of analyses.

We agree that a fraction of Amot-S176E appears to be localized to the PM, while additional protein is found in the cytoplasm and none in the nucleus (see example in Figure 4). In the same figure, YAP is found in all 3 fractions. Importantly, our previous studies (Yi et al., 2013) and those of others suggest that Amot is required for YAP function in the nucleus. Thus, if Amot is the limiting factor and is excluded from the nucleus this should impair the nuclear function of YAP regardless of whether there is “free” YAP elsewhere in the cell. In this model, it is not just simply the exclusion of YAP from the nucleus, but also the exclusion of a limiting factor, namely Amot, that impairs the nuclear function of YAP. In a reciprocal manner, the Amot-S176A mutant is distributed between the nucleus and the cytoplasm and the majority of YAP is nuclear and bound to AmotS176A (Figure 4) and active (Figure 5, Figure 6). Overall this is an excellent point raised by the reviewer and we expanded on this in the discussion of the model in the manuscript. Having said all this, it is indeed quite likely that additional mechanisms are involved in regulation of YAP function.

*9) The authors discuss previous studies showing that AMOT interacts with the actin cytoskeleton and that this interaction is regulated by AMOT phosphorylation. However, they do not address this potentially important pool of AMOT in their study. How does the actin associated pool fractionate in their protocol – with cytoplasmic, nuclear, or membrane fractions?*

We thank the reviewer for this comment. In the revised version of the manuscript we include data assessing the association of the different Amot alleles with F-actin. While most of the other reports relied on co-localization by IF and in vitro studies using purified protein, we wished to directly assess the interaction of Amot with F-actin in cells. To achieve this, we used biotin-XX-phalloidin which preferentially binds F-actin. We then precipitated the F-actin from cells transfected with the different Amot mutants and assessed the levels of Amot that were co-precipitated. This analysis shows that there are no differences in the association of the different mutants with F-actin. While our findings contrast with some of the previous reports regarding the association of Amot with F-actin, we believe that the approach we employed directly assess the interaction between F-actin and Amot, while other reported efforts employed methods that are either indirect (IF) or in vitro. We include this new data in Figure 5 and Discussion.

*10) Figure 5 vs. A and C. The amount of flag PATJ IPed by anti-flag seems extremely low in this panel compared to others, so lack of HA-AMOT S176A co-IPing doesn't seem to be a well-controlled finding.*

We thank the reviewer for catching this. We now provide a revised figure in which the levels of IPed PATJ are similar between the different conditions.

*11) Others have reported that cadherins can also interact with merlin-AMOT complexes at the membrane in addition to PATJ. Do the authors think that this may also be a significant interaction worth examining?*

We thank the reviewer for this comment. We indeed examined this and find that Amot can co-IP with E-cadherin. Moreover, as in the case of PATJ/Pals1 the association is enhanced with the Amot-S176E mutant. We include this new information in Figure 5 and in the text of the manuscript.

*12) The interpretation that the interaction between YAP and Merlin is predominantly occurring in the nucleus and the cytoplasm is supported by the data, but quantification of these experiments would help providing an assessment as to whether this simply reflect the relative levels of YAP across the different fractions or whether there is evidence for a truly differential interaction across fractions. [This would also be consistent with all the results presented for the phosphorylation experiments, so I would suggest considering this possibility in the text]. Perhaps an alternative way to provide support for a compartment-specific interaction would be to employ a fluorescence-based approach (e.g. split GFP, FRET).*

As noted by the reviewer, the data support the formation of the Merlin/YAP/Amot complex predominantly in the cytoplasm and nucleus (Figure 1) and this interaction does not appear to be regulated by phosphorylation status of S176 (Figure 3). We agree that while the data suggests that the relative levels of YAP in the different compartment might underlie the reason for detecting the complex mostly in cytoplasm and nucleus, there could be other factors regulating the interaction. For example, other PTMs on YAP and/or Amot could impact the interaction at the plasma membrane. However, identifying and characterizing these potential events will require an extensive amount of time and effort and is beyond the scope of the current manuscript. As suggested by the reviewer, we included a brief discussion of this possibility in the manuscript.

*13) The presentation of the western blots (IP/western) should be accompanied by quantification. I was surprised that experiments that I would assume should be loaded on the same SDS page gel (e.g. different constructs for the rescue experiments in Figure 3) were shown as separate panels. In the absence of quantification, this makes it very difficult to comment on whether a reduction/increase in complex formation occurs following mutation at this site (though clearly, the interaction is not abrogated by these mutations).*

We agree that it would have been beneficial to have all the samples run on the same gel. However, given the number of samples and the need for clear boundaries between the lanes we had to run the samples on separate gels. Regardless, western blotting using film is only semi-quantitative and therefore for relying on relatively small changes in band intensity, such as those referred to in Figure 3, is problematic. Thus, although we cannot exclude the possibility of minor changes to the affinity of the complex, as the reviewer noted the interaction between Merlin and YAP is not abrogated by the changes in status of Amot-S176. This point does not really alter our interpretation of the results. Further detailed analysis using purified proteins and assessment of complex affinity by sophisticated biochemical approaches will be needed to further resolve this issue.

*14) What is the intracellular localization of endogenous Merlin (and YAP)? The localization experiments currently presented all depend upon transient transfection.*

The localization of endogenous Merlin and Amot can be seen in the IF experiments shown in Figure 4.

*15) For Figure 2, please provide supporting evidence for efficient Amot KD (if the same results have been also obtained with a different Amot hairpin, please also add the results in supplementary to confirm the specificity of the effect on complex formation).*

We provide additional support in Figure 2—figure supplement 1. In this experiment, Amot was knocked down using smartpool siRNA and the effects on Amot levels and the requirement for Amot for the association of Merlin and YAP was confirmed.

*16) If the phosphospecific antibody works in IF, it would be nice to provide additional evidence that the phosphorylated form is enriched in the plasma membrane fraction (alternatively, this could be done by immunoblotting). This would support the results obtained with the S to E mutation.*

Unfortunately, we were unable to optimize the antibody for IF and therefore unable to carry out this experiment.

*17) What is the relative level of expression (in relation to the endogenous) of the different Amot constructs? Similarly, the authors may want to comment on the expression levels of the other constructs they used in relation to the respective endogenous proteins. Could some of the results observed due to overexpression / tagging artifacts?*

Please see our response to question 1.

*18) Concerning the Figure 3—figure supplement 2 (as per Discussion), since a YAP protein in which the 5 serines are mutated to alanines predominantly resides in the nucleus, could it be that the nuclear sequestration is the primary cause of the absence of complex formation?*

Please see our response to question 5.

*19) Other comments: – this likely happened through PDF generation during submission, but the quality of the images is problematic. Labels do not display clearly, and microscopy images are extremely difficult to evaluate. The authors need to provide high quality images with the revision.*

Thank you for point this out. We redid the images at a higher resolution.